# Genome-wide association study meta-analysis of blood pressure traits and hypertension in sub-Saharan African populations: an AWI-Gen study

Surina Singh [1,2] ✉, Ananyo Choudhury [1], Scott Hazelhurst [1,3], Nigel J. Crowther [4], Palwendé R. Boua [1,5], Hermann Sorgho[5], Godfred Agongo[6,7], Engelbert A. Nonterah [7,8], Lisa K. Micklesfield [9], Shane A. Norris [9,10], Isaac Kisiangani [11], Shukri Mohamed [11], Francesc X. Gómez-Olivé [12], Stephen M. Tollman [12], Solomon Choma[13], J-T. Brandenburg[1,14,15] & Michèle Ramsay [1,2,15] ✉

Most hypertension-related genome-wide association studies (GWASs) focus on non-African populations, despite hypertension (a major risk factor for cardiovascular disease) being highly prevalent in Africa. The AWI-Gen study GWAS meta-analysis for blood pressure (BP)-related traits (systolic and diastolic BP, pulse pressure, mean-arterial pressure and hypertension) from three sub-Saharan African geographic regions (N = 10,775), identifies two novel genome-wide significant signals (p < 5E-08): systolic BP near *P2RY1* (rs77846204; intergenic variant, p = 4.95E-08) and pulse pressure near *LINC01256* (rs80141533; intergenic variant, p = 1.76E-08). No genome-wide signals are detected for the AWI-Gen GWAS meta-analysis with previous African-ancestry GWASs (UK Biobank (African), Uganda Genome Resource). Suggestive signals (p < 5E-06) are observed for all traits, with 29 SNPs associating with more than one trait and several replicating known associations. Polygenic risk scores (PRSs) developed from studies on different ancestries have limited transferability, with multi-ancestry PRS providing better prediction. This study provides insights into the genetics of BP variation in African populations.

Hypertension (HTN) is a major risk factor for cardiovascular diseases (CVD) such as coronary heart disease, heart valve diseases, atrial fibrillation, aortic syndromes, cerebral stroke and renal failure[1,2]. Between 1990 and 2019, HTN prevalence almost doubled for adults (aged 30–79 years) and in 2022 affected 1.25 billion people living in low- and middle-income countries[3]. This increase is attributed to population growth and ageing and is predicted to increase to 1.56 billion people by 2025[4,5]. In addition, HTN is a leading risk factor

for premature deaths and disability worldwide[6,7], accounting for 17.9 million deaths in 2018[8]. It is present in ~22% of the global population, with the highest prevalence observed in Africa (27%), particularly in urban communities and in older people[9].

HTN prevalence and awareness differ between and within sub-Saharan African countries[10]. There is a paucity of data on the prevalence, treatment and control of HTN in many African countries and therefore its contribution to related conditions, such as hypertrophic

cardiomyopathy, is not fully understood[11,12]. Major research focuses on genetic associations with HTN, due to its high prevalence and the fact that it doubles the risk for CVD[8,13]. Familial studies have shown HTN associations amongst immediate family members, with genetic factors explaining approximately 30–50% of blood pressure (BP) variation amongst individuals[14,15]. However, these studies have limitations in identifying genetic variants responsible for the increased risk of developing HTN.

Genome-wide association studies (GWASs) have explained 27% of the genetic heritability for BP[16]. The GWAS Catalog[17,18] includes data from the first BP/HTN case-control studies conducted in 2007 for HTN[19] and BP as a quantitative trait[20]. The GWAS Catalog currently includes several thousand independent genetic associations with BP-related traits, based on 380 studies and 586 associations with HTN based on 120 studies (https://www.ebi.ac.uk/gwas, accessed 17 November 2022).

Early GWASs outlined the complexity of studying BP-related traits and emphasized the importance of large sample sizes to enable the detection of genetic associations[19,20]. Large-scale GWAS discovery meta-analyses have shown significant genetic associations with BP and HTN[16,21,22]. The largest BP GWAS to date by Evangelou et al.[16] included over 1 million individuals of European ancestry from the UK Biobank (UKBB) and the International Consortium of Blood Pressure (ICBP), identifying over 1000 independent genetic signals (535 novel) with BP-related loci.

Only a small number of GWAS for genetic associations with BP and HTN have been performed on the African continent. Despite HTN being highly prevalent in Africa[23], most studies have focused on European populations[21]. Studies on African-ancestry populations include mainly African American (AA) populations[24–28], with the first GWAS for HTN in AA conducted in 2009 by Adeyemo et al.[29] Hendry et al.[30] studied a black South African population (n = 1947 with ~700 women who are also present in our study) with samples genotyped using the Metabochip (~200,000 single nucleotide polymorphisms (SNPs) previously associated with cardiometabolic traits). They found genetic associations with systolic and diastolic BP in genes of interest (NOS1AP, MYRF and POC1B) and in some intergenic regions (DACH1|LOC440145 and INTS10|LPL) [30]f.

African populations have high genetic diversity, allele frequency differences and low linkage disequilibrium (LD) when compared to other populations[31] and therefore GWASs from sub-Saharan Africa have the potential to discover novel BP-related SNPs. However, it is important to recognize and adjust for extensive population structure across different African regions[32,33] in genetic association studies and to use genotyping arrays, such as the Human Heredity and Health in Africa (H3Africa) SNP array, that is enriched for common genetic variants in African populations[34].

In this study, the sub-Saharan African cohort of older adult participants referred to as the Africa Wits-INDEPTH partnership for Genomic Studies (AWI-Gen)[35,36], was used and DNA samples were genotyped with the H3Africa SNP array (Fig. 1). The study aimed to identify genetic associations with four continuous BP-related traits (systolic BP (SBP), diastolic BP (DBP) and mean-arterial pressure (MAP)) and one categorical trait (HTN), in three sub-Saharan African regions represented in the AWI-Gen study (Supplementary Data 1). To boost power, the findings were meta-analyzed with other studies that included African or African-ancestry participants. Fine-mapping, genetic risk score analysis and transferability were also assessed.

## Results

Participants in the AWI-Gen cohort had a mean age (SD) of 51.8 (8.2) years, with more women (54.7%) (Table 1). The average BMI of the cohort was 25.1 (6.7) kg/m$^2$ (defined as overweight (body mass index (BMI) between 25.0 and 29.9 kg/m$^2$). The average resting heart rate was within the normal range (<100 beats per minute). The majority of study participants fell within the normal to pre-HTN BP category (126.9/ 83 mmHg; 3818 HTN cases, 6918 HTN controls), with most self-reported as not using anti-hypertension medication (AHM) (76.8%) (Supplementary Data 2). Among individuals identified as having HTN, more had stage 1 (17.2%), were not using AHM (16.3%), didn't have parents with HTN (12.7%) and were unaware of their HTN status and not controlling for HTN (11.5%). The discovery GWASs for the five BP traits (SBP, DBP, HTN, PP, and MAP) was conducted on 10,700 sub-Saharan African participants with 13,976,041 SNPs. For each BP-related trait, quality control (QC) was performed and adjustments were made for the use of AHM (Supplementary Fig. 1). The power calculation revealed that the current study has at least 80% power to detect an effect size beta of -0.60 for SNPs with MAF > 0.10 (Supplementary Fig. 2).

### Genetic associations with BP traits

Genetic associations with each of the five BP traits are shown using Miami and Manhattan plots (Fig. 2). Association studies were performed in two stages: Stage 1 – meta-analysis of the GWAS for the three geographic regions represented in the AWI-Gen cohort (N = 10,775); Stage 2 – meta-analysis of Stage 1 with GWASs from other studies on African and African-ancestry populations (UKBB African-ancestry (UKBBa, N = 3058) and Uganda Genome Resource (UGR, N = 6400)). There was no indication of genomic inflation as observed by the QQ-Plots (Supplementary Fig. 3), since the genomic inflation factor (GIF), lambda (λ), was <1.05 for all five BP traits, indicating adequate control for population sub-structure (Supplementary Fig. 4).

Independent GWASs for each AWI-Gen region (East, West, and South) were conducted (Supplementary Fig. 5). Prior to the Stage 1 meta-analyses, genome-wide (GW) associations ($p < 5E{-}08$) for 38 independent SNPs, with 9 SNPs associated with more than one BP trait (referred to as shared SNPs), were found in the three independent AWI-Gen regions (Supplementary Data 3). Thus, 12 signals each for East (2 shared SNPs) and South (4 shared SNPs), and 14 signals for West (2 shared SNPs) were identified. Due to regional differences, the mega-analyses (a single GWAS for the entire AWI-Gen study for each trait), when compared to the AWI-Gen meta-analyses, gave different GW associations with different associations identified across regions (Supplementary Fig. 5). A meta-analysis of the three independent regions GWASs was done using Han and Eskin's random-effects (RE2) model (Stage 1).

**Stage 1 GWAS.** Suggestive associated genomic regions (or loci) ($p < 5E{-}06$) from the Stage 1 discovery GWAS (identified in FUMA), are shown in Supplementary Data 4. Across the five traits, 129 independent genomic regions were identified, with 29 genomic associating with more than one BP-trait (see bold font SNPs in Supplementary Data 6). When comparing the GWAS by region, the replication of suggestive signals ($p < 5E{-}04$) differed across the East, West and South African regions (Supplementary Data 6).

The GW significance threshold ($p < 5E{-}08$) was reached for SBP with rs77846204 (imputed intergenic variant in RP11-38P22.2, $p = 4.95E{-}08$) (Table 2), driven by the West ($p = 4.16E{-}07$) and East ($p = 3.24E{-}04$) AWI-Gen region GWASs (Supplementary Fig. 6, Supplementary Data 6). This SNP was associated with both DBP ($p = 1.66E{-}06$) and MAP ($p = 1.51E{-}07$) (see bold SNPs in Supplementary Data 6) and had a high allele frequency in previous studies[37] for all ancestries (MAF > 0.2, Ancestries: African, Admixed American, East Asian, European, African Americans). GW significance was also reached for PP with rs115808349 (imputed intergenic variant in ELL2P2, $p = 1.76E{-}08$), driven by the East AWI-Gen region ($p = 2.25E{-}05$) (Supplementary Fig. 6, Supplementary Data 6) and had low allele frequency for all ancestries (MAF < 0.005), with the except for African populations (0.05).

Several suggestive independent genomic regions (Supplementary Data 4) were observed across the five BP-related traits (40 SBP, 25 DBP,

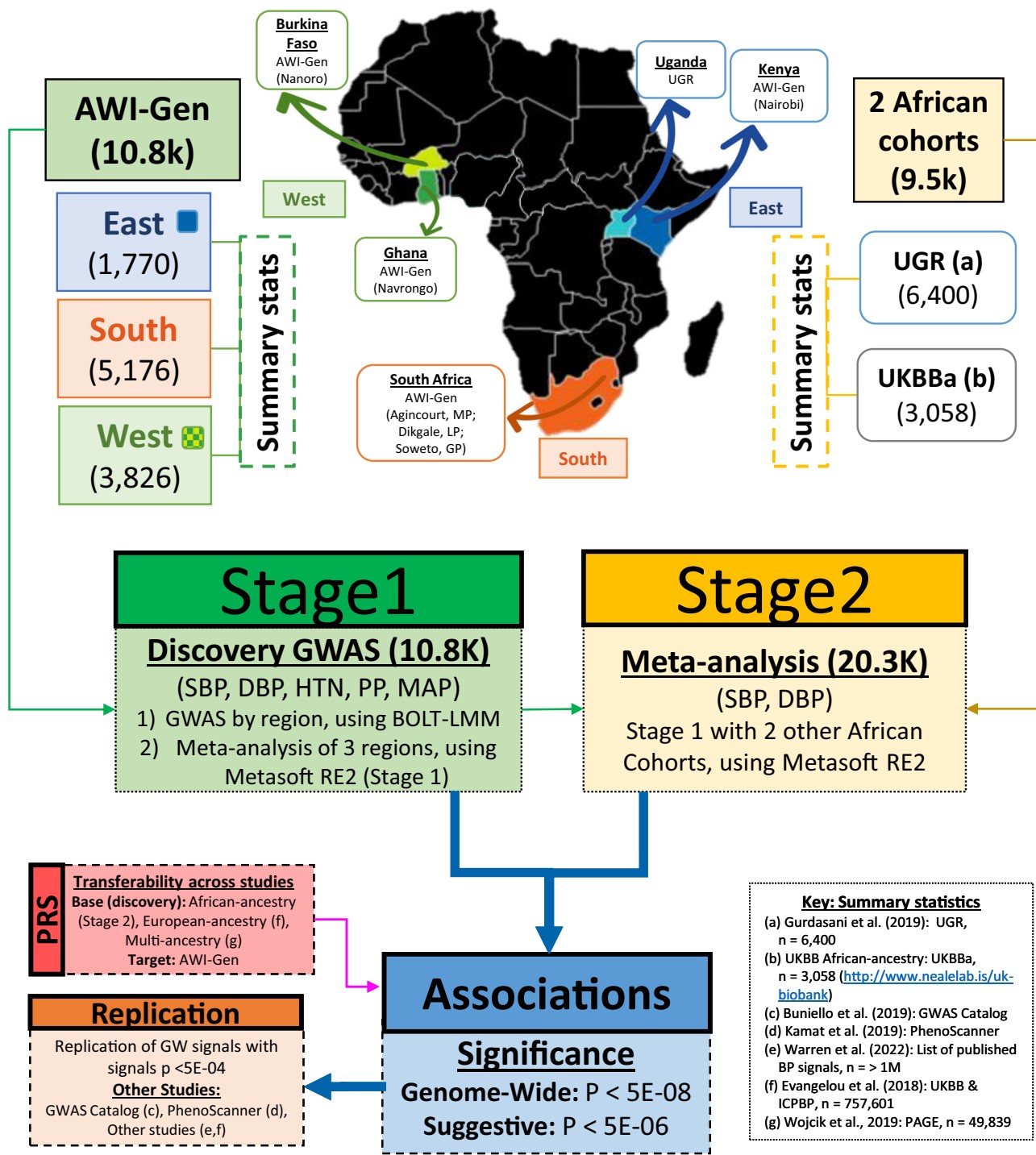

**Fig. 1 | Workflow summary for datasets and analyses.** Stage 1 GWAS was conducted for all five BP-related traits, and a meta-analysis was performed using the summary statistics of the GWAS for each AWI-Gen region (East, South, West) – sample sizes are indicated in brackets. Stage 2 GWAS was conducted for SBP and DBP only, and was a meta-analysis of the Stage 1 results with other African-ancestry summary statistics i.e. UGR[45] and UKBB. Replication of associations was assessed using the GWAS Catalog[17], PhenoScanner[37] and summary statistics from other studies[16,38]. Transferability across studies was conducted via PRS based on African-ancestry (UGR, UKBBa), Multi-ancestry i.e. PAGE[46] and European-ancestry i.e. UKBB & ICBP[16] cohorts (discovery) used to assess the distribution of SBP and DBP risk according to PRS quintiles in the AWI-Gen cohort (target).

21 HTN, 33 PP, and 31 MAP). The strongest signals (lowest $p$-values) that reached suggestive significance were: DBP, rs6494981 (intergenic *TMCO5B*, $p = 9.40E{-}08$), which was also a suggestive signal for MAP ($p = 1.59E{-}06$, shared SNPs); HTN, rs113112741 (intronic *MAML3*, $p = 7.12E{-}08$); MAP, rs73315125 (intergenic *FDPSP6*, $p = 8.02E{-}08$), which was also a suggestive signal for SBP ($p = 6.35E{-}08$) and DBP ($p = 1.13E{-}06$, shared SNPs).

**Stage 2 GWAS**. The number of SNPs included in the analyses increased from 13,952,382 (Stage 1) to 14,845,228 for the Stage 2 GWAS. No GW associations were detected in the Stage 2 analysis for any of the traits. Stage 2 GWAS suggestive independent associated genomic regions ($p < 5E{-}06$), are shown in Supplementary Data 5 and include 40 independent genomic regions (17 SBP, 23 DBP (40 independent SNPs)). Most of these signals were driven by the Uganda Genome Resource (UGR) dataset ($p < 5E{-}04$, Supplementary Data 7). Only one SBP (rs17428471) and five DBP (rs114007149, rs141245590, rs474277, rs617549, rs556594) independent signals were also identified in the Stage 1 GWAS, reaching suggestive significance. The signals with the lowest p-values that reached suggestive significance were: SBP, rs115702999 (ncRNA_exonic *HECW2:AC020571.3*, $p = 2.77E{-}07$); DBP, rs6009081 (intronic *PPARA*, $p = 5.75E{-}07$).

## Replication of Stage 1 and 2 GWAS outcomes

Exact replication was conducted to examine whether any of the current study's GW significant signals ($p < 5E{-}08$) occur at least at a modest replication threshold ($p < 5E{-}04$) (Stage 1 and Stage 2) in any of previous BP GWASs[16,17,37,38]. The absence of replication of the two GW SNPs ($p < 5E{-}08$) for SBP (rs77846204, beta $= -1.99$, $p = 4.95E{-}08$) and PP (rs115808349, beta $= -2.92$, $p = 1.76E{-}08$) in previous BP GWASs ($p < 5E{-}04$) suggests both the signals to be novel to this study.

It was also investigated whether any of the GW significant SNPs detected in previous BP GWASs[16,17,37,38] ($p < 5E{-}08$), showed p-values (with the same beta direction) below this replication threshold ($p < 5E{-}04$) in the current study (Supplementary Data 8). At this threshold, replication of 592 GW significant SNPs, within 131 identified genomic regions ($p < 5E{-}08$) from previous studies were found (500 Stage 1, 115 Stage 2, 23 both Stages). Details of replication for each previous study (i.e. GWAS Catalog[17], Warren et al.[38] and Evangelou et al.[16]) are reported in Supplementary Data 8. Several SNPs that were associated with more than one BP trait were identified, with most replicated SNPs from European-ancestry studies. Thirteen replicated SNPs for Stage 1 and three replicated SNPs for Stage 2, were from four trans-ethnic studies that included African ancestry participants[25,39–42] and all SNPs (except for rs9821489) were associated with more than one BP trait. One replicated SNP, rs17428471, which replicated for a trans-ethnic (including African-ancestry) study[40] for both stages, also replicated from an African-ancestry study[25].

## Fine-mapping and functional analysis

For SBP, the regional plot (Fig. 3) around *P2RY1* showed that within a 1MB flanking region of rs77846204, there were other SNPs previously reported to be associated with SBP and PP[39,43,44] (Supplementary Data 9). Other SNPs in this region were also associated with CVD-linked traits such as HDL cholesterol, lung function/post-bronchodilator (FEV1, associated with lung function), liver function tests and type 2 diabetes. Five SNPs from this region were included in the 95% credible set and the lead SNP (rs73022036) also showed the highest probability of being the causal SNP (logbf > 2) (Supplementary Data 10).

For PP, though the regional plot around *LINCO1256* (rs115808349) (Fig. 3) shows a second peak with rs62317311 (chr 4), that reached suggestive significance ($p = 8.92E{-}07$), only the lead SNP was included in the credible set (Supplementary Data 10). Markers within a 1MB flanking region of rs115808349, were previously associated with resistance to AHM in HTN (Supplementary Data 9), as well as traits

**Table 1 | AWI-Gen study characteristics ($N = 10775$)**

| Characteristics | Number of participants ($n$) | Mean (SD)/Percentage (%) |
|---|---|---|
| Age (years) | 10775 | 51.8 (8.2) |
| Sex (%) | | |
| Female | 5892 | 54.7 |
| Male | 4883 | 45.3 |
| Participants: Site by region | | |
| East | 1771 | 16.4 |
| Nairobi | 1771 | 16.4 |
| South | 5177 | 48.0 |
| Agincourt | 2253 | 20.9 |
| Dikgale | 1143 | 10.6 |
| Soweto | 1781 | 16.5 |
| West | 3827 | 35.5 |
| Nanoro | 1983 | 18.4 |
| Navrongo | 1844 | 17.1 |
| SBP (mmHg) | 10707 | 126.9 (22.6) |
| DBP (mmHg) | 10709 | 83 (16.7) |
| PP (mmHg)[a] | 10703 | 43.9 (12.3) |
| MAP (mmHg)[a] | 10703 | 97.7 (18.0) |
| HTN Status[a,b] | 10708 | |
| Controls (No HTN) | 7634 | 71.3 |
| Cases (HTN) | 3074 | 28.7 |
| BP Status[b] | 10708 | |
| Hypotension (Low BP) | 512 | 4.8 |
| Normal BP | 3724 | 34.6 |
| Pre-HTN | 3398 | 31.5 |
| Stage 1 HTN | 1850 | 17.2 |
| Stage 2 HTN | 1224 | 11.4 |

[a]Based on systolic blood pressure (SBP) and diastolic blood pressure (DBP) values.
[b]Reported hypertension (HTN) status that was adjusted for the use of anti-hypertension medication (AHM).

such as total PHF−tau (SNP × SNP interaction), protein quantitative trait loci (liver) and mood-related traits. Five SNPs from this region were included in the 95% credible set and the lead SNP (rs115808349) also showed the highest probability of being the causal SNP (logbf > 2) (Supplementary Data 10).

Regional plots for the top signal (lowest $p$-value) that reached suggestive significance of association for BP traits in the Stage 1 and 2 GWASs ($p < 5E{-}06$) are shown in Supplementary Fig. 7.

Functional mapping of position, eQTL (matched cis-eQTL SNPs) and chromatin interaction (i.e. 3D DNA−DNA interactions) are reported in Supplementary Data 11 and Supplementary Data 12. The two intergenic SNPs with GW significance had no predicted functional impact.

## PRS

Polygenic risk scores (PRSs), developed from three ancestries (African[45], European[16] and multi-ancestry[46]) GWASs (discovery) were applied to the individuals in the AWI-Gen cohort (target, $N = 10,676$) for SBP and DBP (shown in Fig. 4). Due to a lack of comparative data, this could not be performed for the other three BP-related traits analyzed in this study.

All PRSs developed from studies in the different ancestries, showed an increase in predicting higher BP levels as the quintile scores increased (Fig. 4a). The highest change in effect size (mm/Hg) was observed in the PRS from the multi-ancestry population, whereas the lowest change was observed in the African-ancestry PRS derived from the UKBBa dataset for both SBP and DBP.

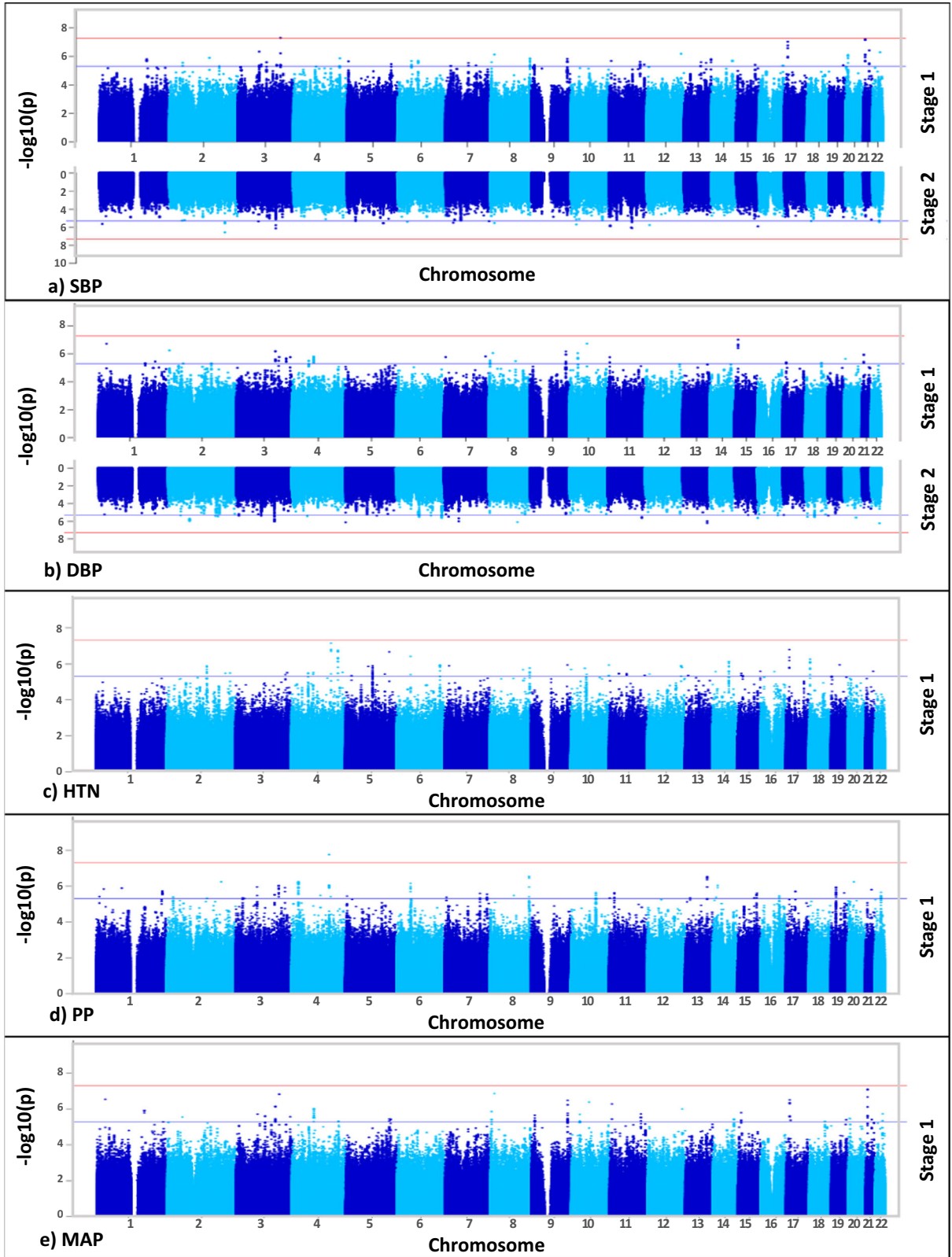

**Fig. 2 | Discovery GWAS genetic associations in AWI-Gen (Stage 1) and the meta-analysis (Stage 2).** Miami plots with results from Stage 1 and Stage 2 are shown for (**a**) SBP and (**b**) DBP and Manhattan plots for Stage 1 for (**c**) HTN, (**d**) PP and (**e**) MAP. GWAS included adjusting for age, age2, sex and the first 10 PCs as covariates. With GW significance = $p < 5E-08$. Miami and Manhattan plot shows $-log10$-transformed two-tailed $p$-value for each BP trait (y-axis) and base pair positions along the chromosomes (x-axis); red line = GW significance ($p < 5E-08$); purple line = threshold for suggestive association ($p < 1E-06$).

**Table 2 | Genome-wide associations (*p* < 5E−08) from Stage 1 AWI-Gen GWAS for BP traits**

| Trait | Lead SNP | CHR:POS | Nearest gene[a] | Type | EA | RA | EAF | BETA | SE | *P*-value |
|-------|----------|---------|-----------------|------|-----|-----|-----|------|-----|-----------|
| SBP | rs77846204 | 3:152582544 | *RP11-38P22.2* | intergenic | A | C | 0.19 | −1.99 | 1.24 | 4.95E−08 |
| PP | rs115808349 | 4:133259109 | *ELL2P2* | intergenic | T | C | 0.05 | −2.92 | 0.85 | 1.76E−08 |

Genome-wide associated signals (*p* < 5E−08) from the Stage 1 discovery GWAS.
Position is given for Build 37 (GRCh37/hg19).
Beta, SE: EAF and *P*-value (two-tailed) calculated using RE2 (Han and Eskin's random-effects) model implemented in METASOFT v2.0.1[70].
*P*-value, Stage 1 meta-analysis *P*-value for RE2 (Han and Eskin's random-effects).
Beta and SE for Random Effects (RE) as provided by METASOFT v2.0.1[70].
*CHR* chromosomes, *POS* gene position, *SNP* single nucleotide polymorphism, *EA* effect allele, *RA* reference allele, *EAF* effect allele frequency, *Beta* effect size estimates for continuous traits i.e. systolic blood pressure (SBP) and pulse pressure (PP), *SE* standard error.
[a]Nearest annotated gene(s) given.

The variance explained between phenotype and risk score estimated, using adjusted R2 i.e. R2 (%), was highest for the multi-ancestry PRS (0.22% for SBP and 0.36% for DBP) and lowest for the African-ancestry PRSs for both SBP (for UKBBa: 0.07%) and DBP (for UGR: 0.04%) (Fig. 4b, Supplementary Data 13). The PRSs generated for the different ancestries (discovery database, with AWI-Gen as the target database), were significant (*p* < *P*-value threshold (PT)) for SBP in UGR African-ancestry, for DBP in European-ancestry and both SBP and DBP in UKBBa and multi-ancestry PRS, indicating transferability (see Supplementary Data 13). The multi-ancestry PRS had the highest number of SNPs for SBP (326,601 SNPs) and the second highest for DBP (69,071 SNPs). The predictivity of the PRS for SBP and DBP, using AUROC (Area Under the Receiver Operating Characteristic curve) and AUC (under the ROC Curve) metrics, suggested statistical significance at the 95% confidence interval (AUC lower bound >0.5) i.e. the ability to accurately distinguish patients with and without elevated BP, except for the UKBBa (AUC = 0.5, lower-upper bond = 0.484−0.51) and Evangelou (AUC = 0.51, lower-upper bond = 0.497−0.522) discovery datasets for SBP (Supplementary Data 13).

## Discussion

HTN is a complex multifactorial disease that involves interactions of multiple variants in many genes, together with environmental risk factors, thereby making the identification of genetic associations complicated. AWI-Gen is a population-based cross-sectional cohort from Africa with a high prevalence of HTN, showing low awareness and control of high BP, suggesting a lack of effective treatment[10].

The strengths of this study include following the same standard procedures and analysis parameters for all AWI-Gen participants across the different geographic regions of Africa and the same genotype array and imputation panels, for a GWAS performed in a population with a high prevalence of HTN. The AWI-Gen discovery GWAS is based on participants from three different African regions that exhibit significant population structure, requiring adjustments during the analysis. To address this limitation, three region-based GWASs were conducted for the East, South and West African regions and meta-analyzed for the AWI-Gen dataset (Stage 1 GWAS). The dataset was of modest sample size, with limited power to identify associations with markers with low minor allele frequency and small effect sizes. To address this limitation, at least partially, the AWI-Gen data was meta-analyzed with published summary statistics from GWASs performed on African and African-ancestry cohorts. The availability of suitable African datasets to replicate the novel associations was a major challenge. Therefore, replication was conducted using both modest African datasets and large European datasets. Finally, the comprehensive evaluation of risk models was dependent on the availability of suitable independent African datasets, with the UGR cohort[45] and UKBBa being the most closely related datasets available. Several suggestive signals (*p* < 5E−06) were identified in the Stage 1 and 2 GWAS (Supplementary Data 4), with the identification of two novel SNPs that reached GW significance for SBP and PP in the Stage 1 discovery GWAS (Table 2).

Regional GW associations across Africa were observed in this study (Supplementary Data 3).

Li et al.[47], based on a machine-learning algorithm to predict new HTN-related genes, predicted the *P2RY1* gene. This gene harbors our novel SBP GW signal (rs77846204; *p* = 4.925E−08), to be one of the top 20 possible HTN genes (Posterior Probability = 0.9750)[41]. This SNP was not GW significant in the fixed effects (FE) model (*p* = 2.58E−05) (Supplementary Data 4, Supplementary Data 6, see Supplementary Note 2.1) due to the variability of effect between regions (Supplementary Fig. 6). Though, indirectly this supports a possible functional connection between the *P2RY1* gene and the trait. Similarly, Sung et al., 2019[48], based on a gene-interaction analysis of smoking with PP and MAP traits in multi-ancestry populations, identified rs147998309 (chr4:133596832) to be associated with PP and current smoking status in African-ancestry. This SNP is located within 300 kb of our GW significant association for PP rs80141533 (near *LINCO1256*, *p* = E−08). As these SNPs are not in LD, we expect them to be novel. Nevertheless, this detection of association in the same genomic region and strengthens the possibility of involvement of this gene/genomic region in PP. The rs115808349 SNP also reached GW significance for the FE model (*p* = 1.25E−08) (Supplementary Data 6, see Supplementary Note 2.1). The lack of multiple large African-ancestry datasets and diversity within each African region could contribute to the lack of replication of the meta-analysis with larger African-ancestry populations. Currently, most African GWASs are limited to cohorts from Uganda, Nigeria and South Africa and studies that include admixed AA populations[49]. Similar to this study, genetic associations linked with BP-related traits in African populations have been found to be limited to those populations[25,26,49,50]. There is a need for larger GWAS of continental African populations, to better investigate the role of these SNPs in BP regulation among Africans.

In contrast to this study's discovery GWAS, the GWASs of both the UKBBa and UGR[45] cohorts did not make adjustments for AHM. The UGR cohort also did not include the first 10 principal components (PCs) as covariates and performed an inverse normal transformation. Other potential African studies had to be excluded, due to the lack of data availability, diversity of populations or pooled admixed African-ancestry datasets[49]. Some datasets had large proportions of participants with traits that could potentially influence HTN status and were therefore excluded from the meta-analyses (these included the Africa America Diabetes Mellitus Study (AADM), Durban Case Control Study (DCC) and Durban Diabetes Study (DDS) cohorts each with ~50% diabetic participants[45]). Having diabetes could affect BP, since it causes the walls of the blood vessels to stiffen which could lead to HTN, with many studies reporting a correlation between diabetes and HTN[51–53]. The lack of knowledge of related HTN co-factors, such as the prevalence and medication used for diabetes not being recorded, is cause for concern when trying to determine genetic association with HTN[51]. Failure to replicate SNPs that reached GW significance in this study could also be due to small sample sizes, which would affect the power of each study to detect associations. In addition, protocols for

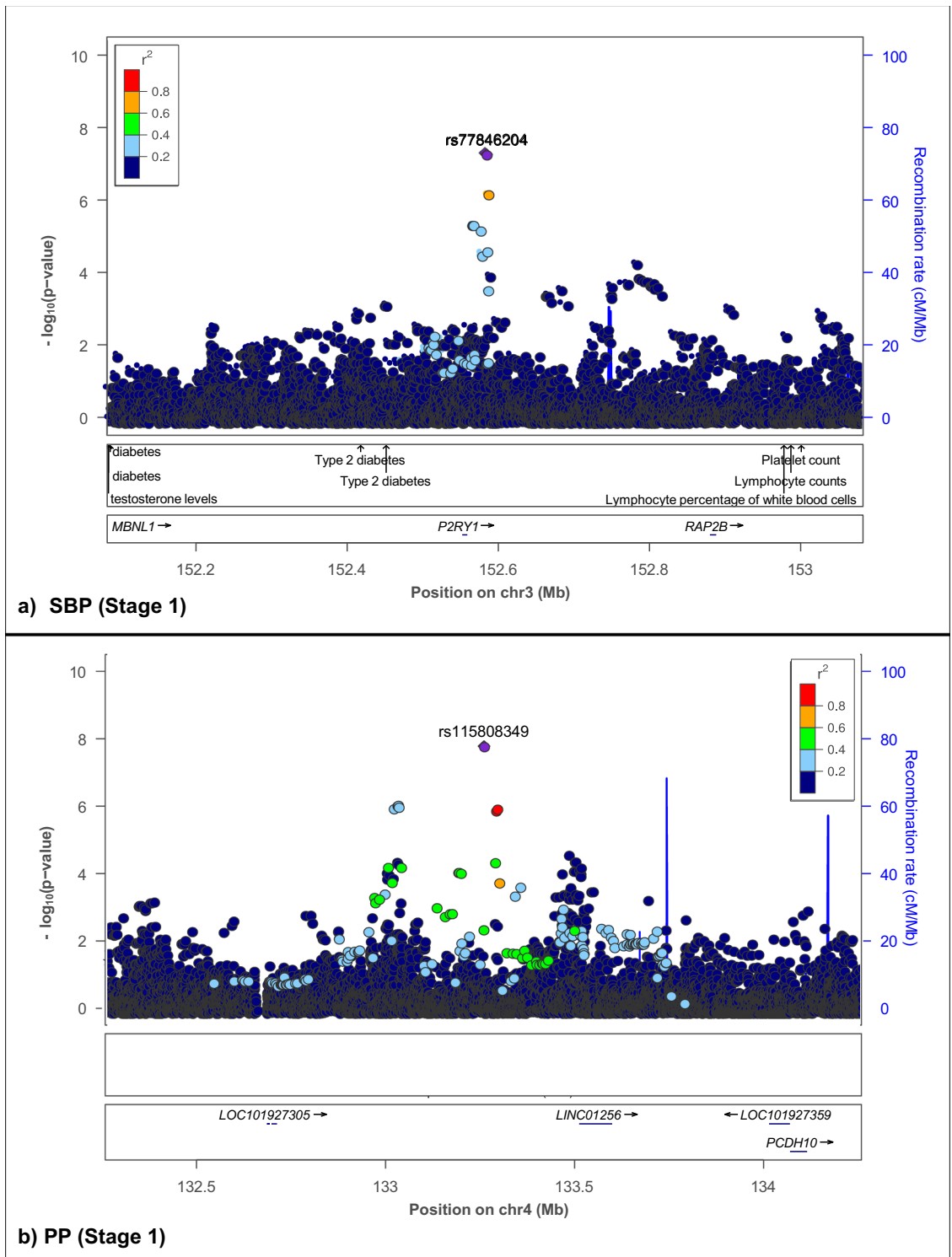

**Fig. 3 | Regional plots of novel GW significant (*p* < 5E−08) associations.**
LocusZoom plots showing GW significant associations for **a** SBP and **b** PP in the
AWI-Gen study. Regional visualization of associated SNP regions was performed
using LD from AWI-Gen in LocusZoom V0.4.8[75], using a 1MB flanking region, which
was compared to data in the GWAS Catalog 17. Lead SNPs are indicated by the
purple diamond and GWAS Catalog trait labels and genes are shown below the
plots. Plots shown for **a** SBP around the *P2RY1* region (rs77846204, *p* = 4.95E−08
and **b** PP around the *LINC01256* region (rs115808348, *p* = 1.76E−08, intergenic
*ELL2P2* – also consisting of rs62317311 (*p* = 8.92E−07), for the AWI-Gen Stage 1
GWAS (*N* = 10,775).

phenotype (measurement errors) and genotype (array and imputation
panels used) data, allele frequency, LD (variants with the causal SNPs)
and effect sizes (attributed gene-environmental interactions) may also
differ[45,49]. The low replication rate of GW associations (Supplementary
Data 8) found in European studies may be attributed to the low power

of the small studies to detect small effect variants. The smaller sample
size for African-ancestry studies is only powered to detect large effect
associations[49].

Similar to this study, several BP-related traits are associated with
the same SNPs, and are mapped to non-coding genomic regions,

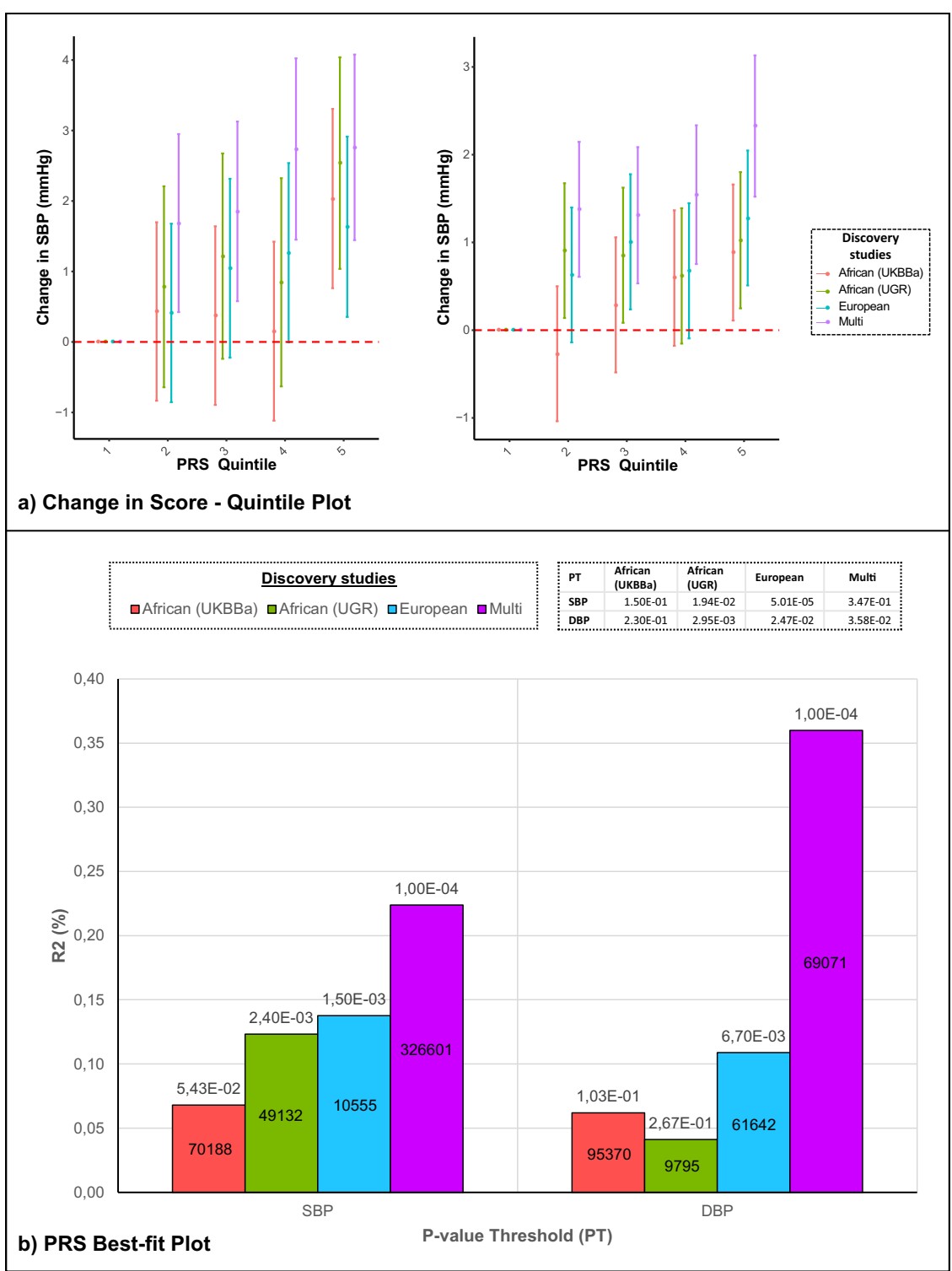

**Fig. 4 | Transferability of Polygenic Risk Score (PRS).** PRSs derived from four GWASs (discovery studies) and applied to the AWI-Gen cohort (target), using PRSice-2 V2.3.5[78]: (1) African-ancestry cohort, UKBBa ($n = 3058$) (2) African cohort, UGR ($n = 6400$)[45] (3) European cohort, UK biobank and ICBP ($N = 757,601$)[16,46] and (4) Multi-ancestry cohort, PAGE ($N = 49,839$ with 17,152 African-ancestry)[40,46]. **a** PRS stratification of SBP and DBP in the AWI-Gen target population: Point range-plots comparing the difference in BP-trait mean (mmHg) of the upper PRS quintiles from the lowest, stratified by the discovery datasets (error bars = mean ± 95% confidence intervals) are shown. **b** Plots showing adjusted variance explained (% R2), between phenotype and risk score estimated, by each PRS for SBP and DBP: *P*-value (above bar, two-tailed) and number of SNPs (within bar) are stated for the *P*-value threshold (PT).

making functional mapping challenging[54]. Blood pressure multi-trait analyses, using the SHet and SHom approaches (which account for the correlation of the multi-traits and overlapping or related samples among the cohorts), provide greater power to detect BP associations with the SHet method resulting in more associations compared to SHom[26,55]. Blood pressure-related traits are closely linked with other CVD risk factors such as insulin resistance, obesity, kidney function, atherogenic dyslipidemia, stroke and coronary artery disease[56].

Heterogeneous regional patterns and associations have been identified between HTN and obesity within continental Africans[57]. The influence of these CVD risk factors on BP-related traits, along with other gene-gene, gene-environment, and lifestyle effects among different geographic regions within Africa, could limit the relatively small AWI-Gen sample population's ability to accurately identify genetic associations and should be further investigated.

Since an individual's genetic data is stable throughout life, it can be used to potentially predict future disease risk. The PRSs derived from multi-ancestry populations provided higher risk prediction of BP-related traits in the AWI-Gen cohort compared to PRSs from other African ancestry populations (Fig. 4). In contrast, PRSs derived from African ancestry populations had higher risk prediction for lipid traits in sub-Saharan Africans compared to European and multi-ancestry scores. This shows that BP-related traits are more complex to understand and the low transferability of PRS to Africans could be attributed to the smaller sample size, differences in LD, allele frequency and gene-environmental factors. The lack of predictivity of current PRSs for key cardiometabolic traits such as HTN highlights the urgent need for additional data and efforts to build larger African-based PRSs. Poor transferability has been demonstrated for existing European-based PRS to African ancestry populations for most phenotypes[4,58–61]. This stresses the importance of further research to optimize PRS prediction in non-European populations, specifically African populations, with increased sample sizes to enhance PRS prediction[58,60]. Machine learning, using GWAS summary statistics, was found to improve the risk prediction of PRSs for traits that influence the risk of HTN (diabetes, obesity/BMI and height) and this suggests that it may also enhance PRSs for hypertension[62,63]. The lack of interpretability of machine learning encourages the development of hybrid techniques, such as combining Mendelian randomization with machine learning post-GWAS approaches, to identify causal inferences of associated variants[64,65].

Future GWASs should focus on regional differences in continental Africa with large sample sizes per region to better understand these associations. The SNPs that were associated with multiple BP-related traits should be explored further, along with conducting BP multi-trait analyses to increase study power[26,55]. Gene–gene and gene–environment interactions of BP-related traits should also be explored to better understand genetic heritability (where only 3–6% can currently be explained by GWASs)[21]. Pharmacogenomics studies that focus on drug–gene interactions and treatment outcomes may lead to improved clinical treatment guidelines[54].

In conclusion, two signals of GW significance were observed in the AWI-Gen GWAS for SBP and PP (Stage 1), with no GW significant associations detected in a meta-analysis with other African-ancestry studies (Stage 2). Several suggestive signals were observed for all traits in analyses for both Stages 1 and 2, with 29 SNPs associated with more than one trait, and several replicating known associations. Limited transferability was observed from PRSs developed from studies in different ancestries, with the best prediction using a multi-ancestry PRS. The identification of new genetic associations with BP-related traits will contribute to understanding the genetic etiology of BP variation in African populations and could help provide additional biological insights.

## Methods
The summary workflow is shown in Fig. 1.

### Ethics statement and consent
Ethical approval was obtained from the Human Research Ethics Committee (HREC) (Medical) of the University of the Witwatersrand (Protocol Number: M190927). This was a sub-study to the AWI-Gen study (Protocol Numbers: M121029, M170880, M2210108). Each of the participating sites also obtained ethics approval from their respective ethics committees. AWI-Gen sample data was used as permitted by the informed consent provided by the study participants and according to the H3Africa policies and guidelines (www.h3africa.org).

### Study participants
Participants were from the AWI-Gen study (10,775 participants, with the majority (89.3%) aged between 40–60 years), located in three African regions (East, West, South) from six sites within four countries (Fig. 1) i.e. East - Kenya (Nairobi); West - Burkina Faso (Nanoro) and Ghana (Navrongo); and South - South Africa (Agincourt, Dikgale, Soweto). Exclusion criteria for the study were: pregnant women, close relatives of existing participants (first and second-degree relatives), recent immigrants (who migrated <10 years ago into the region) and individuals with physical impairments preventing measurement of BP. Further study cohort details can be found in Ramsay et al.[35] and Ali et al.[36].

Singh et al.[49] was used as a reference to identify previous GWAS for BP-related traits in African populations, identifying only one study with summary statistics for SBP and DBP in a continental African population i.e. Gurdasani et al.[45].

### BP measurements
The outcome variables were SBP, DBP, HTN, PP and MAP. A digital sphygmomanometer (Omron M6, Omron, Kyoto, Japan) was used for BP measurements, which were taken three times at 2-minute intervals, with the last two measurements used to calculate the average SBP and DBP levels. PP and MAP were measured and calculated as continuous traits i.e. PP was calculated as the difference between the SBP and DBP and MAP was calculated as the sum of DBP and a third of the PP. HTN (binary trait) was classified according to the Seventh Report of the Joint National Committee on Prevention, Detection, Evaluation, and Treatment of High BP (JNC7) guidelines[66] (see Supplementary Data 1).

Hypertensive individuals (3683 cases; 6018 controls) were defined by the following conditions: individuals previously diagnosed with HTN and/or individuals taking medication for HTN and/or individuals having either SBP equal/above 140 mmHg or DBP equal/above 90 mmHg[10]. Adjustments were made for those taking AHM where 15 mmHg were added to SBP and 10 mmHg were added to DBP ($N = 2293$), as done in previous African studies[26,39,40].

QC was performed on the phenotype data using Stata V15 (StataCorp, College Station, Texas, 77845, US)[67] to assess outliers and distribution (Supplementary Fig. 1). The Winsorise very extreme value approach was used to assess outliers i.e. the values should be <6 standard deviations (SD) above or below the mean, but no such values were observed (Supplementary Fig. 1).

### Genetic data and imputation
**QC.** Genotype data of ~11,000 samples was generated on the 2.3 M SNP H3Africa genotyping array designed to include common African variants (https://chipinfo.h3abionet.org). The H3ABioNet/H3Agwas QC pipeline workflow[68] (https://github.com/h3abionet/h3agwas/tree/master/qc) was used to conduct QC analysis as previously described[58] (see Supplementary Note 1). After QC ~1.7 million SNPs and 10,903 samples remained.

**Imputation.** Genotype imputation was conducted to increase the coverage of genomic variation and allow fine-mapping. The African Genome Resources reference panel (EAGLE2 + PBWT pipeline) at the Sanger Imputation Server (https://imputation.sanger.ac.uk) was used for genotype imputation to increase the coverage of the genome, to narrow down the location of potential causal variants and to capture most haplotype blocks. Post-imputation QC (i.e. removal of indels, rare SNPs) resulted in 13,976,041 SNPs (MAF > 0.01 and info score > 0.6)[58].

Only participants with good-quality phenotype and genotype data were used for the GWAS analyses (N = 10,775). The genome assembly (base pair position) was the GRCh37/hg19.

### Genetic association analysis

The discovery GWAS for the BP-related traits was conducted in two stages (Fig. 1).

**Discovery GWAS (Stage 1 AWI-Gen GWAS).** Potential confounders used as covariates in the GWAS were examined for significance by running a general linear model, using STATA V15[67]. As the participants originate from East, West and Southern Africa, there was significant population structure across regions (Supplementary Fig. 4); moreover, the preliminary analysis indicated relatedness among individuals from some of the AWI-Gen cohorts. Therefore, adjustments based on PCs (addressing genetic population structure) and kinship-matrix (addressing relatedness) were used as covariates. Previously defined confounders were also used as covariates, using Singh et al.[49] as a guideline to determine adjustments (except for BMI which was not adjusted for in the previous studies which were included in the Stage 2 GWAS). All genetic association tests were adjusted for the covariates: age, $age^2$, sex and the first 10 PCs (population structure and geographic region-based adjustments).

The H3ABioNet/H3Agwas Association pipeline workflow[68] was used to conduct the discovery GWAS (https://github.com/h3abionet/h3agwas/). Novel associations were defined using the GWAS significant threshold of $p < 5E{-}08$, with a suggestive threshold of $p < 5E{-}06$. Linear mixed models (LMMs) were used to account for random effects for relatedness. Matrix LMMs were run to test for genetic associations, for an additive genetic model, for four continuous BP traits (SBP, DBP, PP and MAP) and one binary trait (HTN), using the Bayesian LMM association testing approach in BOLT-LMM V2.3.2 mixed model association testing[69]. This approach accounts for relatedness, ancestral heterogeneity (in samples) and any other unaccounted structure within the data.

Independent GWASs for each AWI-Gen region (East, West, South) were conducted and a meta-analysis of summary statistics was conducted, using Han and Eskin's random-effects (RE2) model (see Supplementary Note 2.1), in METASOFT v2.0.1[70]. This was implemented in H3ABioNet/H3Agwas Meta-analysis pipeline workflow[68] (http://github.com/h3abionet/h3agwas/meta/meta.nf), to evaluate the robustness of associations detected in a joint analysis of the AWI-Gen dataset (Stage 1 GWAS).

A power calculation was conducted (study design = continuous trait, independent individuals, hypothesis = gene-interaction, fixed number of samples = 10,903), using Quanto V1.2.3[71]. A graph for power versus effect size (beta) at different allele frequencies (see Supplementary Fig. 2) was constructed in R[72].

**Meta-analysis (Stage 2 GWAS).** Previous studies, including only African and African-ancestry participants were used for a meta-analysis (sub-population of African participants from the UKBB (https://biobank.ctsu.ox.ac.uk) and UGR[45]), to combine with the Stage 1 AWI-Gen meta-analysis GWAS, to improve study power (Stage 2). Permission was obtained to access the genotype and phenotype dataset of the UKBB (research project number: 63215). The UKBBa (N = 3060) was previously QCed and imputed, and a discovery GWAS was conducted, following the same methodology used for the Stage 1 GWAS. Gurdasani, et al.[45] consisted of four African-ancestry cohorts: UGR (N = 6400), DDS (N = 1165), DCC (N = 1542) and AADM (N = 5231). Diabetes causes the walls of the blood vessels to stiffen, which leads to high BP[51–53], therefore AADM, DDS and DCC, which included ~50% diabetic participants, were excluded.

The Stage 2 meta-analysis was conducted for SBP and DBP by comparing the Stage 1 GWAS (meta-analysis of AWI-Gen GWAS by region) with the UKBBa dataset (N = 3058) and the UGR cohort (N = 6400)[45] summary statistics, following the same methodology used for the Stage 1 meta-analysis. Other BP-related traits (HTN, PP, and MAP) could not be included due to the lack of data availability in cohorts used in the Stage 2 GWAS.

**Visualization and interpretation of genetic associations.** Miami plots were generated, to display significantly associated SNPs in associated regions, using the qqman package in R[73]. Genomic control (λ) was evaluated in R[72] and quantile-quantile (Q-Q) plots were constructed in FUMA[74] as a QC check, to re-evaluate genetic inflation and confounding biases such as cryptic relatedness and population stratification (with the assumption that the regional groupings will be independent of each other – see Supplementary Note 2.2). Several SNPs were identified, that were associated with more than one BP-related trait, meeting suggestive significance ($p < 5E{-}06$) in each trait.

Regional visualization of associated SNP regions was performed using LD from AWI-Gen in LocusZoom V0.4.8[75], using a 1MB flanking region, which was compared to data in the GWAS Catalog[17].

### Replication with previous findings

Replication of the Stage 1 and 2 GWAS with populations of similar genetic ancestry was performed, using the exact replication strategy[76]. Replication was also tested against the Stage 1 and 2 GWAS, using previous studies: (1) GWAS Catalog[17]; (2) PhenoScanner[37]; (3) List of all 3800 published BP-associated SNPs; (4) European-only ancestry population from Evangelou et al.[16], consisting of the UKBB & ICBP cohorts (currently the largest published study with 757,601 European ancestry individuals). Warren et al.[38] reported only genome-wide SNPs ($p < 5E{-}08$) and summary statistics were not available. Therefore, replication of suggestive SNPs ($p < 5E{-}04$) could not be assessed for all studies. With the availability of summary statics for Evangelou, et al.[16], it was possible to include the replication of suggestive SNPs for bi-directional replication analysis. Multiple rows of duplicate SNPs have been included, since the same SNP was found to replicate ($p < 5E{-}04$) for more than one trait and/or GW significant, for more than one previous study. Any duplicate signals from the same study across the previous study databases were removed.

Replication of the Stage 1 and 2 GWAS with previous studies, was assessed by comparing GW associations ($p < 5E{-}08$) against SNPs with suggestive associations ($p < 5E{-}04$). In addition, replication of GW signals found in previous studies ($p < 5E{-}08$) were compared against the Stage 1 and 2 GWAS suggestive associations ($p < 5E{-}04$).

The H3ABioNet/H3Agwas Replication pipeline workflow was implemented to conduct replication analysis (https://github.com/h3abionet/h3agwas/tree/master/replication). The exact replication method was used to test for replication with the GWAS Catalog (see Supplementary Note 2.3). (https://www.ebi.ac.uk/gwas/, accessed on 27 March 2022).

The GW associations ($p < 5E{-}08$) were also compared against the PhenoScanner database[37], since the GWAS Catalog is limited to GW signals with $p < 5E{-}08$ with a few suggestive at $p < 5E{-}06$, to pick up any missed or additional suggestive signals ($p < 5E{-}04$) found within this database (http://www.phenoscanner.medschl.cam.ac.uk). Replication of the Stage 1 and 2 GWAS was also compared against a list of 3800 published BP-associated SNPs listed within Warren, et al.[38]

Exact replication was tested by using Evangelou et al. (2018) summary full statistics data (currently the largest published study with 757,601 European ancestry individuals), for SBP, DBP and PP, to determine which signals were uniquely identified in studies with African-ancestry populations. Since both genome assemblies were GRCh37/hg19, a direct comparison was evaluated in R[72]. The different regions of the AWI-Gen datasets were also compared for South, East and West Africa (which were found to be significantly different sub-populations).

Novel-associated SNPs were determined by searching for SNPs within a 500 kb region of all SNPs with GW associations (*p*-value < 5E−08) found in our study.

## In silico functional analysis

FUMA[74] was used for in silico functional analysis and annotation was performed to select the most likely causal variants from the GWAS summary statistics. The FUMA pipeline (https://fuma.ctglab.nl) was used for functional gene mapping, using the SNP2GENE tool, for positional, expression quantitative trait loci (eQTL) and chromatin interaction mappings. Candidate SNPs were selected in the associated genomic regions with R2 ≥ 0.6 to define independent significant SNPs with GW (*p* < 5E−08) and MAF ≥ 0.01 for annotation (reference population = 1000 G Phase3 African; included variants in reference panel (non-GWAS tagged SNPs in LD); maximum distance between LD blocks to merge into a locus = 250 kb). Candidate SNPs functional consequences were predicted by chromosome base-pair position, and reference and alternate alleles, to databases containing known functional annotations (see Supplementary Note 2.4).

## Fine-mapping

Fine-mapping to identify potential causal variants was conducted by comparing the GW associations and/or top association SNPs found in Stage 1 and 2 GWAS, with previously reported BP loci. Lead SNPs were defined as SNPs within a genomic region that had the lowest *p*-value (per BP-trait) i.e. potential causal variants (at 95% confidence interval). The H3ABioNet/H3Agwas Finemaping pipeline workflow[68] was used for fine-mapping (https://github.com/h3abionet/h3agwas/tree/master/finemapping), to identify potential causal variants and credible sets (region set at 300 kb, using *p*-value z-scores to re-estimate beta and se). Shogun stochastic search was performed to identify credible sets of potential causal variants, at a 95% confidence level, using FINEMAP V1.0[77] which employs Bayesian calculation of posterior probability.

## Polygenic risk score (PRS)

The PRS analysis was conducted using trait-specific effect-weighted variants obtained from the discovery GWAS, using PRSice-2 V2.3.5[78] (see note Supplementary Note 3.1).

The AWI-Gen study genotype data was used as the target database and could only be applied to SBP and DBP (due to BP-related trait data availability) from previous study groups by ancestry. Previous studies included: (1) African-ancestry only: A meta-analysis was conducted with the UKBBa dataset (*N* = 3058) and the UGR cohort (*N* = 6400)[45] summary statistics, using the Han and Eskin's random-effects (RE2) model in METASOFT v2.0.1[70], for the PRS African-ancestry only dataset; (2) European-ancestry only: Evangelou et al.[16] summary statistics, which consisted of UKBB and ICBP cohorts (*N* = 757,601), was used for the PRS European-ancestry only dataset; (3) Wojcik, et al.[46] summary statistics, which consisted of the Population Architecture using Genomics and Epidemiology (PAGE) (*N* = 49,839 non-European individuals with 17,152 AA) cohort, was used for the overall European multi-ancestry dataset (AA, Hispanic/Latino, Asian-ancestry, Native Hawaiian-ancestry, Native American-ancestry).

The prediction models were adjusted for the same covariates used for the GWAS analysis i.e. age, age2, sex and the first 10 PCs, generated within PRSice-2 V2.3.5[78]. The adjusted R-squared (adj-Rsq) was used to account for predictors that are not significant in a regression model. The Adj-Rsq was computed using residuals after adjustment (adj-Rsq). The best predictive PRS were estimated using the highest adj-Rsq. The *P*-value threshold (PT) was determined in PRSice-2 V2.3.5[78], by calculating the empirical *P*-value for each PRS (algorithms described in Supplementary Note 3.3). Different PTs were identified for each trait and compared using Rsq, where the best PT was defined by the highest Rsq.

In addition, AUROC and AUC metrics were conducted using the pROC[79] package in R[72], to evaluate the performance and reclassification of a PRS model for the risk prediction. The risk stratification of PRSs was evaluated using quintile plots (comparing the difference in the mean of the phenotypic trait between the upper and lowest quintile. When AUC (Area under the ROC Curve) lower bound >0.5, it suggests statistical significance (with a 95% confidence interval) i.e. the ability to accurately diagnose patients with and without elevated SBP and DBP based on the test.

## Reporting summary

Further information on research design is available in the Nature Portfolio Reporting Summary linked to this article.

## Data availability

The AWI-Gen data set is available from the European Genome-phenome Archive (EGA) database (https://ega-archive.org/), with accession number EGAS00001002482 (phenotype dataset: EGAD00001006425; genotype dataset: EGAD00010001996, genome assembly: GRCh37/hg19). The availability of these datasets is subject to controlled access through, the Data and Biospecimen Access Committee of the H3Africa Consortium. The processed data generated in this study are provided in Supplementary Material. The summary statistics reported in the paper are accessible on the GWAS Catalog (https://www.ebi.ac.uk/gwas/). Permission was obtained to access the genotype and phenotype dataset for UKBB (research project number: 63215) (as described in Methods). Publicly available databases include (1) GWAS Catalog[17] (https://www.ebi.ac.uk/gwas/; BP, EFO_0004325; SBP, EFO_0006335; DBP, EFO_0006336; HTN, EFO_0000537; PP, EFO_0005763; MAP: EFO_0006340), (2) PhenoScanner[37] (http://www.phenoscanner.medschl.cam.ac.uk/, Traits: BP, SBP, DBP, HTN, PP, MAP). Other summary statistics reported in the paper are accessible on the GWAS Catalog (https://www.ebi.ac.uk/gwas/) for (1) Gurdasani, et al.[45] African-ancestry UGR cohort (SBP: GCST009053, DBP: GCST009052). (2) Evangelou, et al.[16] European-ancestry UKBB & ICBP cohorts (SBP: GCST006624, DBP: GCST006630, PP: GCST006629). (3) Wojcik, et al.[46] multi-ancestry PAGE cohort (SBP: GCST008044, DBP: GCST008029). Source data are provided as a Source Data file. Source data are provided with this paper.

## Code availability

The H3ABioNet/H3Agwas GWAS pipeline workflow[68,80] was employed for QC, association testing, meta-analysis and fine-mapping (as described in the methods section, available at https://github.com/h3abionet/h3agwas).

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

## Acknowledgements

The authors would like to acknowledge the AWI-Gen field workers, phlebotomists, laboratory scientists, administrators, data personnel and all other staff who contributed to the data and sample collections, processing, storage, and shipping, and the participants without whom this work would not have been possible. The AWI-Gen Collaborative Centre is funded by the National Human Genome Research Institute (NHGRI), Office of the Director (OD), Eunice Kennedy Shriver National Institute Of Child Health & Human Development (NICHD), the National Institute of Environmental Health Sciences (NIEHS), the Office of AIDS Research (OAR) and the National Institute of Diabetes and Digestive and Kidney Diseases (NIDDK), of the National Institutes of Health (NIH, grant number: U54HG006938) and its supplements, as part of the H3Africa Consortium. Additional funding was granted by the Department of Science and Technology (now Department of Science and Innovation), South Africa (award number: DST/CON 0056/2014). S.S. and J-T.B. were supported by the South African National Research Foundation (NRF) through MR's South African Research Chair in Genomics and Bioinformatics of African populations hosted by the University of the Witwatersrand (SARChI), funded by the Department of Science and Technology, and administered by the NRF. J-T.B. is also funded by the Cancer Association of South Africa and the Science for African Foundation - REACCT-CAN Grant (Del-22-008). The views expressed in this manuscript do not necessarily reflect the views of the funder. This research has been conducted using the UK Biobank Resource under Application Number 63215.

## Author contributions

The study was designed by S.S., J-T.B., A.C. and M.R. S.S. and J-T.B. performed the analysis. S.H., N.C., P.R.B., H.S., G.A., E.A.N., L.K.M., S.A.N., I.K., S.M., F.X.G-O., S.M.T. and S.C. directed the field work and sample collection. S.S. drafted the first draft and incorporated feedback and suggestions for further analyses from M.R., J-T.B. and A.C. All authors read and approved the submitted version.

## Competing interests

The authors declare no competing interests.

## Additional information

[1]Sydney Brenner Institute for Molecular Bioscience, Faculty of Health Sciences, University of the Witwatersrand, Johannesburg, South Africa. [2]Division of Human Genetics, National Health Laboratory Service and School of Pathology, Faculty of Health Sciences, University of the Witwatersrand, Johannesburg, South Africa. [3]School of Electrical and Information Engineering, University of the Witwatersrand, Johannesburg, South Africa. [4]Department of Chemical Pathology, National Health Laboratory Service, Faculty of Health Sciences, University of the Witwatersrand, Johannesburg, South Africa. [5]Clinical Research Unit of Nanoro, Institut de Recherche en Sciences de la Sante, Ouagadougou, Burkina Faso. [6]Department of Biochemistry and Forensic Sciences, School of Chemical and Biochemical Sciences, C.K. Tedam University of Technology and Applied Sciences, Navrongo, Ghana. [7]Navrongo Health Research Centre, Ghana Health Service, Navrongo, Ghana. [8]Julius Global Health, Julius Centre for Health Sciences and Primary Care, University Medical Centre Utrecht, Utrecht, Netherlands. [9]SAMRC Developmental Pathways for Health Research Unit, Faculty of Health Sciences, University of the Witwatersrand, Johannesburg, South Africa. [10]School of Health and Human Development, University of Southampton, Southampton, UK. [11]African Population and Health Research Center, Nairobi, Kenya. [12]MRC/Wits Rural Public Health and Health Transitions Research Unit (Agincourt), School of Public Health, Faculty of Health Sciences, University of the Witwatersrand, Johannesburg, South Africa. [13]Department of Medical Science, Public Health and Health Promotion, School of Health Care Sciences, Faculty of Health Sciences, University of Limpopo, Polokwane, South Africa. [14]Strengthening Oncology Services, Faculty of Health Sciences, University of the Witwatersrand, Johannesburg, South Africa. [15]These authors contributed equally: J-T. Brandenburg, Michèle Ramsay. ✉e-mail: surinsingh@hotmail.co.za; Michele.Ramsay@wits.ac.za

