## [Peer Review File · Nature Communications]

Genome-wide association study meta-analysis of blood pressure traits and hypertension in sub-Saharan African populations: an AWI-Gen studyREVIEWER COMMENTS

Reviewer #1 (Remarks to the Author):

Singh and colleagues present a study with the aim to identify genetic variation associated with blood pressure traits and hypertension in a Sub-Saharan African Population. The topic is of importance. As outlined by the authors, hypertension is a serious and undertreated condition in particular in Africa. As I am not an expert in statistics, I limit my comments to the interpretation of the results.

The most prominent result is the identification of two novel blood variants which are associated with systolic blood pressure (rs77846204) and pulse pressure (rs80141533) at the genome-wide level of statistical significance. Unfortunately, these associations could not be replicated in a meta-analysis with published studies including individuals from African populations.

1. I am not persuaded that the associations are true because both SNPs are very rare (rs77846204 basically only exists in African populations; rs80141533 has a MAF of 0.02 in Africans) and results may be false-positive due to population stratification. As a result, there is no real peak of association signals in the Locuszoom plots. This, overall, renders this and all subsequent analysis somewhat futile.
2. Fig. 1 should be simplified and maybe moved to the Supplemental Material.
3. Fig. 1 should be moved to the Supplemental Material.
4. In the tables, only the genome-wide significant results should be shown. Table 4 should be moved to the Supplemental Material.

Reviewer #2 (Remarks to the Author):

This paper makes a valued contribution to the field, by performing a GWAS in African Ancestry (discovery $N \sim 10k$), showing the importance of research in diverse ancestry, specifically showing the diversity within African Ancestry, and the differences between African-American data vs data on Africans actually within Africa. Due to the differences within African regions, I commend the authors for performing a random-effects meta-analysis for combining their data. As the authors state, HTN has highest prevalence in African populations, so an African BP-GWAS is indeed important.

The authors perform a comprehensive GWAS project, analysing 5 different BP related traits, performing 2-stage meta-analysis, and some several secondary analyses, such as PRS, and bi-directional lookups comparing results with lots of other publicly available GWAS results datasets.

Their main results are the identification of 2 genome-wide significant signals in their discovery GWAS.

Specific Comments & Qs to Authors:

- 1) The Abstract does not state if the 2 GW-significant signals are "novel". As a Reviewer, I can confirm that both are novel. So I agree with the authors conclusion on line 174. But, I think this conclusion of novelty should be emphasized more, for clear reporting. Having checked the BP-GWAS literature, I note that the PP top SNP is $\sim 300kb$ from SNP rs147998309, chr4:133596832, which was reported within the non-EUR (I believe AA) sample from the GxL analysis of smoking for PP & MAP traits by Sung et al, 2019 (PMID: 31127295), but they are not in LD, so still novel signals, but interesting for the authors to note.

2) I am not keen on the authors' use of the term "pleiotropic" referring to a SNP being associated with more than 1 of the 5 BP traits. Usually, I believe pleiotropy would be interpreted to be broader than this, so if I were commenting on pleiotropy of a BP-associated SNP, I would be referring to associations with other traits and diseases, that are not BP traits. I think the authors are over-stating pleiotropy here. I agree it is nice to show whether SNPs show association in more than 1 of the 5 GWAS trait analyses, and this may add evidence of robustness for some of the loci discovered. But the idea of pleiotropy here of the SNPs is over-estimated, and given too much detail and weight within the manuscript.

3) I think the term "Fine-mapping" has also been over-stated. Essentially, in the Figures, the authors have provided Locus-Zoom plots for the top signals. But fine mapping usually refers to a much more extensive, rigorous analytical pipeline than this, using lots of conditional analyses, Bayesian analyses, bioinformatics analyses, etc. So I think this is therefore mis-leading.

4) The authors have done a great amount of work for the PRS analyses section, and produce great Figures. However, the actual statistical results that they provide seem quite limited, simply the p-value and the R-sq for % Variance Explained, in addition to the quantile plot figures. Could they consider other metrics too, e.g. AUROC; NRI for reclassification, etc? And could the SBP and DBP PRS be used to evaluate prediction of HTN in the African individuals?

5) Please can I query the Rsq and %VE result values at line ~225. Firstly, do you mean 0.22%, or 22%? Secondly, I am surprised by high the 2 values are for the multi-ancestry PRS, compared to the other PRS. Please can I check that the authors have used adj-Rsq rather than Rsq? And that authors are only calculating Rsq for the residual trait, after eliminating variation from sex, age, PC covariates, etc?

6) Please restrict the number of different p-value significance thresholds that you use, and be more consistent with the thresholds you use. In the primary analysis, it is nice and clear, that $P < 5e-8$ refers to GW-significance, and $P < 5e-6$ for suggestive significance, which is fine. But then, in addition to this, you also seem to use $P < 1e-6$ for suggestive significance too on line 137; $P < 5e-4$ for suggestive significance on line 141; $P < 5e-4$ for replication suggestive significance on line 177. These latter ones are confusing.

7) Please make sure that all numbers counting significant variants etc, can be easily followed through the manuscript. E.g. line 126 states 41 SNPs, but later I count up $12 + 12 + 14 = 38$ instead. Similarly, lines 138-139 it is confusing to interpret the different numbers output from FUMA, with 129 independent regions / 136 independent SNPs / 130 lead SNPs, without extra clarity and definition from the authors.

8) Furthermore, are these 41 SNPs independent, or are some within the same loci? Please also make sure there is distinct clarity between SNP vs loci, etc.

9) I note that the authors have referred to the list of 3,800 published BP associations from Warren et al (2022). As the authors have accessed this pre-print paper to use this list, I am surprised that the authors did not also use this latest $N > 1M$ BP-GWAS pre-print paper to aid their analyses in other parts of the manuscript. For example, in addition to using the list of 3,800 published BP SNPs, did the authors also check their 2 significant loci against the 113 novel loci identified in this pre-print, or include these 113 novel loci within their bidirectional replication lookups?

Furthermore, the authors seem to have done extra work to generate their own PRS from the Evangelou et al 2018 GWAS summary stats using PRSice, whereas the Warren et al (2022) pre-print also includes data for an already generated EUR BP-PRS for SBP, DBP and PP.

10) In the paragraph lines 126-135, the phrasing "with only XX reaching suggestive significance"...please make this clearer? Do you mean that the GWAS has 6 additional loci reaching suggestive significance, i.e. in addition to the ones stated to have been GW-significant? Or, are you comparing to suggestive significance in the meta-analysis?

11) I find the whole paragraph from lines 175-185 rather confusing and hard to follow, please re-write this more clearly.

12) I also got confused with lack of clarity in the 2nd half of the PRS results paragraph, lines 127-130. In line 227, rather than "The PRS...", please state exactly which PRS you are referring to. I am also confused when the authors state e.g. "in UGR" and "in European-ancestry", because I had understood that PRS had been generated by the EUR GWAS summary stats data, for example, and then tested "in" the AWI-Gen data.

13) Please re-consider your data availability request: "All data that support the findings of this study are available from the corresponding authors on request." >> It is important these days that all GWAS summary stats data are publicly deposited for all published studies. Please confirm in your rebuttal where your data will be made publicly available.

Tables & Figures:

1) Please simply re-title the Locus Zoom plots as "Locus Zoom Plots" rather than Fine Mapping figures (See comment above)

2) In Fig 2, the QQ plots could be Sup Figs, as these relate more to QC. Then focus is given to the Miami/Manhattan plots which actually present the results.

3) In Table 1, please define AHM in the Table legend, as well as listing it in the overall manuscript's abbreviations list.

4) Table 2 is too large to be a Main Table. This will need to be Sup. Also, as it shows suggestive signals, it should be Sup. Please only include a smaller Main Table for the 2 novel GW-significant signals. Also, this is an output Table from FUMA, so many of these columns are not necessary, and provide more detail than required.

5) Likewise, Table 3 has too many detailed columns that are not essential. And it should be Sup if it is for suggestive signals.

6) Table 4 is certainly far too large for a Main Table! This will also need to be Sup. Please also make the A1/A2 alleles of the Previous Studies columns consistent with the EA/RA alleles of the GWAS columns, so that there is alignment for the effect allele and corresponding MAF values. Please make it clearer in the legend that the Previous Studies columns are showing MAF values. I am not sure why there need to be multiple row entries for duplicated SNPs in this Table.

Minor Comments:

1) When the authors refer to the PAIGE multi-ancestry PRS data as a multi-ancestry "model", I think the terminology "model" is mis-leading, and implies that extra modelling has been conducted. Please refer to this instead as e.g. the "PRS from the multi-ancestry GWAS". Similarly, for clarity, line 229, "The multi-ancestry models had the highest number of SNPs for SBP (N = 326,601) and second highest for DBP (N = 69071)." >> I am assuming the authors mean that the multi-ancestry "PRS" contains this number of SNPs. But this is not clear, and it could be misleading within a GWAS paper referring to SNP associations.

2) Similarly, I wonder whether "multi-ancestry" is the best terminology for the Wojcik et al PRS from PAIGE? It actually doesn't include EUR at all, whereas most multi-ancestry GWAS are mixed of EUR + non-EUR ancestry. So I think it would be important to clarify that this is a fully non-EUR mixed-ancestry GWAS dataset.

3) In the Introduction, line 68 referring to all the GWAS-catalog associations is mis-leading, saying there are ~8,000 associations. These are not independent signals, as many of these associations are not unique, and relate to correlated SNPs. Also, did the authors restrict their searches of GWAS-Catalog and PhenoScanner to $P < 5e-8$ GW-significance threshold?

4) Typo: "GWAS studies", but the "S" = "studies" (line 71)

5) Line 76 in Intro: For clarity, note that the Evangelou et al 2018 BP-GWAS identified 535 "novel" loci, but in total reported >1,000 independent genetic signals.

6) Typo: line 98, "the" should be "that"

7) Line 147: please provide REFs for "previous studies" with regards to other MAFs.

8) Typo: GWAS "Catalog" not "Catalogue"

9) Line 187: please state the p-value threshold for the "top signal SNPs"

10) Lines 218-219, please also provide the REFs here for the datasets you used for PRS

11) Please make it clear in your REFs list, if any REFs are pre-prints, e.g. REF 38, perhaps others...

Reviewer #3 (Remarks to the Author):

The authors have performed a GWAS study in an understudied population and should be congratulated. However, there several main and minor comments that needs to be addressed.

Major

The authors perform a meta-analysis using the Han and Eskin's random effects model. The specific model supposed to increase the power of a study under heterogeneity. However, it has been empirically observed that the specific model, many times, provides over-estimated pvalues. I would strongly suggest to run the meta-analysis using conventional random-effects models, as in the vast majority of performed GWASs. Also, the wording at the last sentence of this paragraph seems kind of confusing, not sure which method the authors compare against. Please, also make sure about the jargon used here. RE2 is not a method, it's just how Metasoft describes the Han and Eskin random-effects for simplicity.

Overall the wording in the methods section for stage 1 and 2 is kind of repetitive, try to make this more clear by removing staff to the supplementary material.

I find the design of this study a little bit confusing. So, the researchers perform GWAS and meta-analysis in stage 1 which is fine. Then in the so called stage 2, additional cohorts are added for SBP and DBP only, correct? Therefore, why you choose to present summary results from each stage separately given that for SBP and DBP the sample has been doubled?

I'm confused regarding the results of fine-mapping provided in the manuscript. Based on methods the authors have used a Bayesian approach but in the reporting of the results it seems that they focus mostly of on the regional locus zoom plots which actually is not fine-mapping unless I'm missing something here.

Please report the rules for QC of BP traits to assess outliers. Also, the binary outcomes should be removed from the supplementary figure cited.

How were the close relatives of the participants have been identified and excluded?

Table 2 is huge and not informative as is. Keep only the necessary information if needed and move the rest to the supplementary material. Same for tables 3 and 4, no need to be at the main manuscript.

The PRS analysis is not clear. Which pvalue threshold did the authors choose in order to construct their PRSs? They mention deciles in their analyses but they present quartiles in the figures.

It is highly suggested that the methods should be polished and provide exact information that the reader can go through the manuscript, tables and figures. As it stands right now it seems that in a lot of places, the authors just choose the prespecified thresholds provided from the tools they use without providing this kind of information to the reader.

Minor

The reported heritability estimates in lines 65-66 are for African populations? Is so, please make it clear to the text. If it is a general statement, the heritability explained for blood pressure traits is now quite larger (line 66). The authors can check a paper that they already cite in their manuscript (Evangelou et al.) that provides larger estimates.

Line 96: I would not call HTN a trait, it's a medical condition. Please harmonize through out the text.

Line 98: Please correct "the" to "that"

Line 523: Please change "ICPBP" to "ICBP". Same in line 566

Line 227: Please show the pvalue

Genome-wide Association Study Meta-analysis of Blood Pressure Traits and Hypertension in Sub-Saharan African Populations: An AWI-Gen Study

Responses in blue to reviewer comments:

The Authors thank the reviewers for important feedback and guidance to improve the quality of the manuscript. Major revisions are included, in line with the Reviewer comments and are shown in the text below in bold for easy identification, with tracked changes made in the word/pdf documents and highlights for the excel documents:

- Singh et al. 2023. GWAS Paper_V2_TC: Contains manuscript.
- Singh et al. 2023. Supplementary Information: Contains Supplementary Figures, which were moved from main manuscript (Figure S 1-7) and Supplementary Notes (Supplementary Note 1-3).
- Singh et al. 2023. Supplementary Data: Contains Supplementary Tables, which were moved from main manuscript (Tables S 1-13).

The Comments below reference the updated Version 2. Therefore, line numbers, tables, figures and supplementary data names may have changed based on updates made.

Reviewer #1	2
Reviewer #2	4
Reviewer #3	14
References	19

Reviewer #1

Comments / Concerns

1. I am not persuaded that the associations are true because both SNPs are very rare (rs77846204 basically only exists in African populations; rs80141533 has a MAF of 0.02 in Africans) and results may be false-positive due to population stratification. As a result, there is no real peak of association signals in the Locuszoom plots. This, overall, renders this and all subsequent analysis somewhat futile.

While the reviewer raises a valid concern:

- Population stratification was accounted for in the discovery GWAS using linear mixed models for each of the three AWI-Gen geographic regions (which account for relatedness, ancestral heterogeneity (in samples) and any other unaccounted structure within the data) and adjusting for the first 10 PCs (population structure and geographic region-based adjustments). See as stated in Results - Genetic associations with BP traits:
 - “There was no indication of genomic inflation **as observed by the QQ-Plots (Figure S 3)**, since the genomic inflation factor (GIF), lambda (λ), was <1.05 for all five BP traits, indicating adequate control for population sub-structure (**Figure S 4**).
 - “To account for the diverse signals driven by each independent region (regional differences), a meta-analysis of the three independent region GWASs was done using Han-Eskin random effects (RE2) model (Stage 1)”.
- In addition, using the genome-wide association threshold of $5E-08$ accounts found false positive rate to other multiple testing.
- Regional differences could attribute to the low number of genome-wide association identified.
 - “Several suggestive signals ($p < 5E-06$) were identified in the Stage 1 and 2 GWAS (**Table S 4**), with the identification of two novel SNPs that reached GW significance for SBP **and PP** in the Stage 1 discovery GWAS (**Table 2**). Regional GW associations across Africa were observed in this study (Table S 3).”

To further address this concern, further research was conducted to find supporting evidence for these associations. The following was added to discussion:

- “**Li, et al. ¹, based on a machine-learning algorithm to predict new HTN-related genes, predicted the *P2RY1* gene. This gene harbors our novel SBP GW signal (rs77846204; $p=4.925E-08$), to be one of the top 20 possible HTN Genes (Posterior Probability=0.9750) ².**”
- “**Similarly, Sung et al., 2019 ³, based on a gene-interaction analysis of smoking with PP and MAP traits in multi-ancestry populations, identified rs147998309 (chr4:133596832) to be associated with PP and current smoking status in African-ancestry. This SNP is located within 300kb of our GW significant association for PP rs80141533 (near *LINC01256*, $p=E-08$). As these SNPs are not in LD, we expect this signal (rs80141533) to be novel. Nevertheless, this detection of association in the same genomic region and strengthens the possibility of involvement of this gene/genomic region in PP.**”

2. Fig. 1 should be simplified and maybe moved to the Supplemental Material.

Figure 1 is an important summary of the study design and is usually a main figure for GWAS papers. Due to the complexity of the study design, if simplified further, important aspects of the study design will be lost from the figure. The Authors would therefore motivate that Figure 1 should remain.

3. Fig. 1 should be moved to the Supplemental Material.

This comment is linked to the previous comment (no.2) and the same justification remains.

4. In the tables, only the genome-wide significant results should be shown. Table 4 should be moved to the Supplemental Material.

The reviewer raises a good point, as this table was too long to remain as main table. The following Table modifications were made:

- Table 2 has been modified to only show the two genome-wide signals and the original Table 2 moved to the supplementary table (now referenced as Table S 4). Unnecessary columns removed in both Table 2 and Table S 4.
- Table 3 had unnecessary columns removed (same columns as Table 2). Table 3 moved to the supplementary table (now referenced as Table S 5).
- Table 4 was too large for main tables and move to supplementary material (now referenced as Table S 8).
- There are now two main tables (Table 1-2).

Reviewer #2

Specific Comments & Qs to Authors:

1) The Abstract does not state if the 2 GW-significant signals are “novel”. As a Reviewer, I can confirm that both are novel. So I agree with the authors conclusion on line 174. But, I think this conclusion of novelty should be emphasized more, for clear reporting. Having checked the BP-GWAS literature, I note that the PP top SNP is ~300kb from SNP rs147998309, chr4:133596832, which was reported within the non-EUR (I believe AA) sample from the GxL analysis of smoking for PP & MAP traits by Sung et al, 2019 (PMID: 31127295), but they are not in LD, so still novel signals, but interesting for the authors to note.

The authors would like to thank the reviewer for picking up that the abstract should also state these signals are novel as mentioned in results and providing crucial information to add to this study.

- “Novel” included in Abstract to describe the GW signals
- Included in discussion:
“Sung et al., 2019³, based on a gene-interaction analysis of smoking with PP and MAP traits in multi-ancestry populations, identified rs147998309 (chr4:133596832) to be associated with PP and current smoking status in African-ancestry. This SNP is located within 300kb of our GW significant association for PP rs80141533 (near *LINC01256*, p=E-08). As these SNPs are not in LD, we expect this signal (rs80141533) to be novel. Nevertheless, this detection of association in the same genomic region and strengthens the possibility of involvement of this gene/genomic region in PP.”
- In addition, the following was included in discussion:
“Li, et al.¹, based on a machine-learning algorithm to predict new HTN-related genes, predicted the *P2RY1* gene. This gene harbors our novel SBP GW signal (rs77846204; p=4.925E-08), to be one of the top 20 possible HTN Genes (Posterior Probability=0.9750)².”

2) I am not keen on the authors’ use of the term “pleiotropic” referring to a SNP being associated with more than 1 of the 5 BP traits. Usually, I believe pleiotropy would be interpreted to be broader than this, so if I were commenting on pleiotropy of a BP-associated SNP, I would be referring to associations with other traits and diseases, that are not BP traits. I think the authors are over-stating pleiotropy here. I agree it is nice to show whether SNPs show association in more than 1 of the 5 GWAS trait analyses, and this may add evidence of robustness for some of the loci discovered. But the idea of pleiotropy here of the SNPs is over-estimated, and given too much detail and weight within the manuscript.

The reviewer raises a valid concern with this definition being too broad. “Pleiotropy” is now removed and only mentioned where necessary as SNPs “associated with more than one BP trait” or as “Shared SNPs” when mentioned in brackets.

3) I think the term “Fine-mapping” has also been over-stated. Essentially, in the Figures, the authors have provided Locus-Zoom plots for the top signals. But fine mapping usually refers to a

much more extensive, rigorous analytical pipeline than this, using lots of conditional analyses, Bayesian analyses, bioinformatics analyses, etc. So I think this is therefore mis-leading. Figure 3 has been renamed from “Fine-Mapping” to “Regional plots”. Additional analyses were included for Finemapping:

- **Methods - Fine-mapping (addition in bold):**
- **“The H3ABioNet/H3Agwas Finemapping pipeline workflow was used for fine-mapping (<https://github.com/h3abionet/h3agwas/tree/master/finemapping>), to identify potential causal variants and credible sets (region set at 300kb, using p-value z-scores to re-estimate beta and se). Shogun stochastic search was performed to identify credible sets of potential causal variants, at 95% confidence level, using FINEMAP ⁴ which employs Bayesian calculation of posterior probability.”**
- **Included supplementary data:**
- **“Table S 10: Fine-mapping the region around independent significant SNPs for BP-related traits with GW significance ($p < 5E-08$).”**
- **Results - Fine-mapping and functional analysis:**
- **For SBP: “Five SNPs from this region were included in the 95% credible set and the lead SNP (rs77846204) also showed highest probability of being the causal SNP ($\log_{bf} > 2$) (Table S 10).”**
- **For PP: “Five SNPs from this region were included in the 95% credible set and the lead SNP (rs115808349) also showed highest probability of being the causal SNP ($\log_{bf} > 2$) (Table S 10).”**

4) The authors have done a great amount of work for the PRS analyses section, and produce great Figures. However, the actual statistical results that they provide seem quite limited, simply the p-value and the R-sq for % Variance Explained, in addition to the quantile plot figures. Could they consider other metrics too, e.g. AUROC; NRI for reclassification, etc? And could the SBP and DBP PRS be used to evaluate prediction of HTN in the African individuals?

Additional analyses were included for PRS (AUROC):

- **Methods - Polygenic risk score (PRS):**
- **“In addition, AUROC and AUC metrics were conducted using the pROC ⁷⁸ package in R ⁷¹, to evaluate the performance and reclassification of a PRS model for the risk prediction. The risk stratification of PRSs were evaluated using quintile plots (comparing the difference in the mean of the phenotypic trait between the upper and lowest quintile. When AUC (Area under the ROC Curve) lower bound > 0.5 , it suggests statistical significance (with a 95% confidence interval) i.e. the ability to accurately diagnose patients with and without elevated SBP and DBP based on the test.”**
- **Included supplementary data:**
- **“Table S 13: Predictivity of PRS models in the AWI-Gen cohort.”**
- **Results - PRS:**
- **“The predictivity of the PRS for SBP and DBP, using AUROC (Area Under the Receiver Operating Characteristic curve) and AUC (under the ROC Curve) metrics, suggested statistical significance at the 95% confidence interval (AUC lower bound > 0.5) i.e. the ability to accurately distinguish patients with and without elevated BP, with the exception of the UKBBa (AUC=0.5, lower-upper bond=0.484-0.51) and Evangelou (AUC=0.51, lower-upper bond=0.497-0.522) discovery datasets for SBP (Table S 13).”**
- **Discussion**
- **“The lack of predictivity of current PRSs for key cardiometabolic traits such as HTN highlights the urgent need for additional data and efforts to build larger African-based PRSs”**

5) Please can I query the R^2 and %VE result values at line ~225. Firstly, do you mean 0.22%, or 22%? Secondly, I am surprised by high the 2 values are for the multi-ancestry PRS, compared to the other PRS. Please can I check that the authors have used adj- R^2 rather than R^2 ? And that authors are only calculating R^2 for the residual trait, after eliminating variation from sex, age, PC covariates, etc?

The value is correct at 0.22 % (using adj- R^2 - unfortunately the variance explained is extremely low).

R^2 has been computed using residuals after adjustment (adj- R^2): The adj- R^2 (residuals) was used (using the best-fit PRS output file (“PRSout.best”) generated within PRSice-2, which gives values in regression). Adjustments were made for the same covariates used for the GWAS analysis i.e. “age, Age2, sex, pc1, pc2, pc3, pc4, pc5, pc6, pc7, pc8, pc9, pc10.”

For clarification of R^2 used, the following changes were made:

- Methods - Polygenic risk score (PRS):
 - “The prediction models were adjusted for the same covariates used for the GWAS analysis i.e. age, age2, sex and the first 10 PCs, generated within PRSice-2 V2.3.5 7. The adjusted R-squared (adj- R^2) was used to account for predictors that are not significant in a regression model. The Adj- R^2 was computed using residuals after adjustment (adj- R^2). The best predictive PRS were estimated using highest adj- R^2 . The P-value threshold (PT) was determined in PRSice-2 V2.3.5 7, by calculating the empirical P-value for each PRS (algorithms described in Supplementary Note 3.3). Different PT were identified for each trait and compared using R^2 , where the best PT was defined by the highest R^2 .”
- Included supplementary data (including R^2 and adjusted R^2 values):
 - “**Table S 13: Predictivity of PRS models in the AWI-Gen cohort.**”
- Results - PRS:
 - “The variance explained between phenotype and risk score estimated, using adjusted R^2 i.e. R^2 (%), was highest for the multi-ancestry PRS (0.22% for SBP and 0.36% for DBP) and lowest for the African-ancestry PRSs for both SBP (for UKBBa: 0.07%) and DBP (for UGR: 0.04%) (Figure 4b, **Table S 13**).”

6) Please restrict the number of different p-value significance thresholds that you use, and be more consistent with the thresholds you use. In the primary analysis, it is nice and clear, that $P < 5e-8$ refers to GW-significance, and $P < 5e-6$ for suggestive significance, which is fine. But then, in addition to this, you also seem to use $P < 1e-6$ for suggestive significance too on line 137; $P < 5e-4$ for suggestive significance on line 141; $P < 5e-4$ for replication suggestive significance on line 177. These latter ones are confusing.

The authors would like to thank the reviewer for picking up this inconsistency. The following corrections were made:

- Corrections have been made throughout paper for suggestive significance which should be $P < 5E-06$.
- Replication significance is correct at $P < 5E-04$ as this is generally less stringent than GWAS suggestive significance. When comparing GWAS suggestive signals ($5E-06$) with signals by region (as a form of replication) the signals by region used a replication suggestive significance of $5E-04$. This was not clearly stated in the methods as mentioned by reviewer and has now been corrected to state “**replication suggestive signals**” throughout paper were $5E-04$ was used for consistency.

7) Please make sure that all numbers counting significant variants etc, can be easily followed through the manuscript. E.g. line 126 states 41 SNPs, but later I count up 12 + 12 + 14=38 instead. Similarly, lines 138-139 it is confusing to interpret the different numbers output from FUMA, with 129 independent regions / 136 independent SNPs / 130 lead SNPs, without extra clarity and definition from the authors.

The Authors would like to thank the reviewer for picking up this error. The number of SNPs were rechecked: As per Table S 3, the number of total associations were indeed 38 SNPs for the comparison of regions (excluding the mega-analysis).

FUMA identifies lead SNPs with P-value less than or equal to $P < 5E-06$ and independent from each other. Independent SNPs were incorrectly labeled and should be what was listed as lead SNPs. This was confirmed by running clumping in Plink within the H3Africa pipeline i.e. it was confirmed that the identified independent SNPs were the same as FUMA lead SNPs.

Independent regions are now referred as “**independent loci**”.

The following changes were made:

- Results - Genetic associations with BP traits (Line 126):
 - “Prior to the Stage 1 meta-analyses, genome-wide (GW) associations ($p < 5E-08$) for 38 independent SNPs, with 9 SNPs associated with more than one BP trait (referred to as shared SNPs), were found in the three independent AWI-Gen regions (Table S 3).”
- Results – Stage 1 GWAS (Line 138-139):
 - “Across the five traits, **129 independent genomic regions were identified, with 29 genomic associating with more than one BP-trait** (see bold font SNPs in Table S 6).”

8) Furthermore, are these 41 SNPs independent, or are some within the same loci? Please also make sure there is distinct clarity between SNP vs loci, etc.

The SNPs are independent and are now correctly labeled throughout the paper (as outlined in previous comment).

9) I note that the authors have referred to the list of 3,800 published BP associations from Warren et al (2022). As the authors have accessed this pre-print paper to use this list, I am surprised that the authors did not also use this latest $N > 1M$ BP-GWAS pre-print paper to aid their analyses in other parts of the manuscript. For example, in addition to using the list of 3,800 published BP SNPs, did the authors also check their 2 significant loci against the 113 novel loci identified in this pre-print, or include these 113 novel loci within their bidirectional replication lookups? Furthermore, the authors seem to have done extra work to generate their own PRS from the Evangelou et al 2018 GWAS summary stats using PRSice, whereas the Warren et al (2022) pre-print also includes data for an already generated EUR BP-PRS for SBP, DBP and PP. Warren, et al. ⁸ (pre-print), was also used for exact replication (see Methods- Replication with previous findings & Results -Replication of Stage 1 and 2 GWAS outcomes). Warren, et al. ⁸ reported only genome-wide SNPs ($p < 5E-08$) and full summary statistics were not available. Therefore replication suggestive SNPs ($p < 5E-04$) were not reported for all reported studies including Evangelou, et al. ⁹. With the availability of summary static for Evangelou, et al. ⁹,

which is the largest BP GWAS study with summary statistics available, it was made it possible to include these replication suggestive SNPs for bi-directional replication analysis.

In addition, with Warren, et al. ⁸ still remaining as a pre-print that has not been peer-reviewed by a journal, it would be risky to request full summary statistics for any additional analyses at this stage.

For clarification, the following changes were made:

- Methods - Replication with previous findings:
 - **“Warren, et al. 8 reported only genome-wide SNPs ($p < 5E-08$) and summary statistics were not available. Therefore, replication of suggestive SNPs ($p < 5E-04$) could not be assessed for all studies. With the availability of summary statistics for Evangelou, et al. 9, it was possible to include the replication of suggestive SNPs for bi-directional replication analysis.”**

10) In the paragraph lines 126-135, the phrasing “with only XX reaching suggestive significance”...please make this clearer? Do you mean that the GWAS has 6 additional loci reaching suggestive significance, i.e. in addition to the ones stated to have been GW-significant? Or, are you comparing to suggestive significance in the meta-analysis?

The sentence “*Thus, 12 signals each for East (2 displaying pleiotropic) and South (4 pleiotropic), with 5 and 11 SNPs reaching suggestive significance ($p < 5E-06$) respectively, were observed. For West Africa 14 signals (2 pleiotropic) with only six reaching suggestive significance (Table S 6) were identified.*”, is a combination of GW regional SNPs Table S 3 and suggestive regional SNPs ($p < 5E-06$) compared with stage 1 meta GWAS suggestive SNPs ($p < 5E-06$) and these values should not be combined. The suggestive SNPs in Table S 6 are relevant to replication suggestive SNPs ($p < 5E-04$) and therefore these values were removed from results.

For clarification, the following changes were made:

- Results - Genetic associations with BP traits (Lines 126-135):
 - “**Thus, 12 signals each for East (2 shared SNPs) and South (4 shared SNPs), and 14 signals for West (2 shared SNPs) were identified.**”

11) I find the whole paragraph from lines 175-185 rather confusing and hard to follow, please re-write this more clearly.

The reference to the number of results across the traits and the traits that had the same SNP with GW significance was report unnecessarily.

There was also an error in reporting the replication results that were corrected. Non-GW significant values from previous studies were removed from Tables. The direction of effect alleles were reported in the opposite direction for some studies, therefore the effect allele was reported in the same direction by changing the EAF (1- reported EAF) and changing the sign of the reported beta values for these specific studies. Therefore, this table was re-evaluated and the updated results were reported.

For clarification, the following changes were made:

- Results - Replication of Stage 1 and 2 GWAS outcomes (Lines 175-185)

- “It was also investigated whether any of the GW significant SNPs detected in previous BP GWASs ⁸⁻¹¹ ($p < 5E-08$), showed p-values (with the same beta direction) below this replication threshold ($p < 5E-04$) in the current study (Table S 8). At this threshold, replication of 592 GW significant SNPs, within 131 identified genomic regions ($p < 5E-08$) from previous studies were found (500 Stage 1, 115 Stage 2, 23 both Stages). Details of replication for each previous study (i.e. GWAS Catalog ¹⁰, and Evangelou, et al. ⁹) is reported in Table S 8. Several SNPs that were associated with more than one BP trait were identified, with most replicated SNPs from European-ancestry studies. Thirteen replicated SNPs for Stage 1 and three replicated SNPs for Stage 2, were from four trans-ethnic studies that included African ancestry participants ^{2,12-15} and all SNPs (with the exception of rs9821489) were associated with more than one BP trait. One replicated SNP, rs17428471, which replicated for a trans-ethnic (including African-ancestry) study ¹⁴ for both stages, also replicated from an African-ancestry study ¹³.”

12) I also got confused with lack of clarity in the 2nd half of the PRS results paragraph, lines 127-130. In line 227, rather than “The PRS...”, please state exactly which PRS you are referring to. I am also confused when the authors state e.g. “in UGR” and “in European-ancestry”, because I had understood that PRS had been generated by the EUR GWAS summary stats data, for example, and then tested “in” the AWI-Gen data. The referenced lines where modifications were made are 227-230 (Results-PRS) instead of 127-130 (Results-Genetic associations with BP traits).

For clarification, the following changes were made:

- Results - PRS (Lines 227-230)
- “The PRSs generated for the different ancestries (discovery database, with AWI-Gen as the target database), were significant ($p < P$ -value threshold (PT)) for SBP in UGR African-ancestry, for DBP in European-ancestry and for both SBP and DBP in UKBBa and multi-ancestry PRS, indicating transferability (see Table S 13). The multi-ancestry PRS had the highest number of SNPs for SBP (326,601 SNPs) and second highest for DBP (69,071 SNPs).”

13) Please re-consider your data availability request: “All data that support the findings of this study are available from the corresponding authors on request.” >>It is important these days that all GWAS summary stats data are publicly deposited for all published studies. Please confirm in your rebuttal where your data will be made publicly available.

The statement “All data that support the findings of this study are available from the corresponding authors on request.” has been removed from the Data availability section. The data will be added to the GWAS Catalog and made available upon publication: “The summary statistics reported in the paper are accessible on GWAS Catalog (<https://www.ebi.ac.uk/gwas/>).”

The process to submit to the GWAS Catalog has started (Submission ID: 644b91e41286510001cdaba0) and full summary statistics will be uploaded once the paper has been accepted from publication.

Tables & Figures:

1) Please simply re-title the Locus Zoom plots as “Locus Zoom Plots” rather than Fine Mapping figures (See comment above)

Figure 3 has been renamed from “Fine-Mapping” to “Regional plots”. Additional analyses were included for Finemapping (As outlined in Specific Comments).

2) In Fig 2, the QQ plots could be Sup Figs, as these relate more to QC. Then focus is given to the Miami/Manhattan plots which actually present the results.

The QQ plots from Figure 1 were moved to Supplementary information as a new Supplementary Figure: “**Figure S 3: Q–Q plots with the genomic control coefficient (λ) showing the discovery GWAS genetic associations in AWI-Gen (Stage 1) and the meta-analysis (Stage 2) for five BP traits.**”

3) In Table 1, please define AHM in the Table legend, as well as listing it in the overall manuscript’s abbreviations list.

Added to Table 1 legend “* reported HTN status that was adjusted for **the use of anti-hypertension medication (AHM).**”. For consistency, each BP-related trait abbreviation was also defined in Table legends: Systolic blood pressure (SBP), diastolic blood pressure (DBP), hypertension (HTN), pulse pressure (PP) and mean-arterial pressure (MAP).

4) Table 2 is too large to be a Main Table. This will need to be Sup. Also, as it shows suggestive signals, it should be Sup. Please only include a smaller Main Table for the 2 novel GW-significant signals. Also, this is an output Table from FUMA, so many of these columns are not necessary, and provide more detail than required.

Table 2 has been modified to only show the two genome-wide signals and the original Table 2 moved to the supplementary table (now referenced as Table S 4). Unnecessary columns removed in both Table 2 and Table S 4.

5) Likewise, Table 3 has too many detailed columns that are not essential. And it should be Sup if it is for suggestive signals.

Table 3 had unnecessary columns removed (same columns as Table 2). Table 3 moved to the supplementary table (now referenced as Table S 5).

6) Table 4 is certainly far too large for a Main Table! This will also need to be Sup. Please also make the A1/A2 alleles of the Previous Studies columns consistent with the EA/RA alleles of the GWAS columns, so that there is alignment for the effect allele and corresponding MAF values. Please make it clearer in the legend that the Previous Studies columns are showing MAF values. I am not sure why there need to be multiple row entries for duplicated SNPs in this Table.

The reviewer raises a good point, as this table was too long to remain as main table. Table 4 was too large for main tables and move to supplementary material (now referenced as Table S 8).

There was also an error in reporting the replication results that were corrected. Non-GW significant values from previous studies were removed from Tables. The direction of effect alleles were reported in the opposite direction for some studies, therefore the effect allele was reported in the same direction by changing the EAF (1- reported EAF) and changing the sign of

the reported beta values for these specific studies. Therefore, this table was re-evaluated and the updated results were reported (As addressed in Reviewer 2's Specific comment 11).

The legend in Table S 8 has been updated to state "EAF, effect allele frequency **with minor allele frequencies (MAF) reported for previous studies.**"

For clarification of why multiple rows of duplicate SNPs were included, the following was included:

- Methods - Genetic association analysis - Replication with previous findings:
 - **"Multiple rows of duplicate SNPs have been included, since the same SNP was found to replicate ($p < 5E-04$) for more than one trait and/or GW significant, for more than one previous study. Any duplicate signals from the same study across the previous study databases were removed."**

Minor Comments:

1) When the authors refer to the PAIGE multi-ancestry PRS data as a multi-ancestry "model", I think the terminology "model" is mis-leading, and implies that extra modelling has been conducted. Please refer to this instead as e.g. the "PRS from the multi-ancestry GWAS". Similarly, for clarity, line 229, "The multi-ancestry models had the highest number of SNPs for SBP (N=326,601) and second highest for DBP (N=69071)." >>I am assuming the authors mean that the multi-ancestry "PRS" contains this number of SNPs. But this is not clear, and it could be misleading within a GWAS paper referring to SNP associations. When referring to PRS, the word "model" has been replaced with "PRS" throughout the document. "N=" has been removed since no reference is given to what number it refers to and clarified what the number refers to SNPs.

For clarification, the following changes were made:

- Results – PRS (Line 229)
 - "The multi-ancestry **PRS** had the highest number of SNPs for SBP (326,601 **SNPs**) and second highest for DBP (69,071 **SNPs**)".

2) Similarly, I wonder whether "multi-ancestry" is the best terminology for the Wojcik et al PRS from PAIGE? It actually doesn't include EUR at all, whereas most multi-ancestry GWAS are mixed of EUR + non-EUR ancestry. So I think it would be important to clarify that this is a fully non-EUR mixed-ancestry GWAS dataset.

The reviewer is correct; PAGE only contains 49,839 non-European individuals. Clarification has been given in Methods section(Methods Wojcik, et al. ¹⁶ summary statistics, which consisted of the Population Architecture using Genomics and Epidemiology (PAGE) (N=49,839 **non-European individuals** with 17,152 AA) cohort, was used for the overall **non-European** multi-ancestry dataset (AA, Hispanic/Latino, Asian-ancestry, Native Hawaiian-ancestry, Native American-ancestry).

3) In the Introduction, line 68 referring to all the GWAS-catalog associations is mis-leading, saying there are ~8,000 associations. These are not independent signals, as many of these associations are not unique, and relate to correlated SNPs. Also, did the authors restrict their searches of GWAS-Catalog and PhenoScanner to $P < 5 \times 10^{-8}$ GW-significance threshold?

The Authors agree that the associations in the GWAS Catalog are not independent. For clarification, the following modification was made:

- Introduction – PRS (Line 68)
- “The GWAS Catalog currently includes **several 1,000 independent** genetic associations with BP, based on 380 studies and 586 associations with HTN based on 120 studies (<https://www.ebi.ac.uk/gwas>, accessed 17 November 2022).”

Yes, $p < 5 \times 10^{-8}$ was used GW-significance for replication of the Stage 1 and 2 GWAS with previous studies. The error identified with Table S 8 showing non-GW values for previous studies has been corrected as outlined in Reviewer 2’s Specific comment 11.

4) Typo: “GWAS studies”, but the “S”=“studies” (line 71)
Correction done: “GWAS”

5) Line 76 in Intro: For clarity, note that the Evangelou et al 2018 BP-GWAS identified 535 “novel” loci, but in total reported >1,000 independent genetic signals.

Correction done:

“The largest BP GWAS to date by Evangelou, et al.⁹ included over 1 million individuals of European ancestry from the UK Biobank (UKBB) and the International Consortium of Blood Pressure (ICBP), **identifying over 1000 independent genetics signals (535 novel) with BP-related loci.**”

6) Typo: line 98, “the” should be “that”
Correction done: “that”

7) Line 147: please provide REFs for “previous studies” with regards to other MAFs.

Correction done. Previous Studies AF: Extracted from the PhenoScanner database – reference added (also to Table S 6 footer):

“**This SNP associated with both DBP ($p=1.66 \times 10^{-6}$) and MAP ($p=1.51 \times 10^{-7}$) (see bold SNPs in Table S 6) and had a high allele frequency in previous studies¹¹ for all ancestries (MAF>0.2, Ancestries: African, Admixed American, East Asian, European, African Americans).**”

8) Typo: GWAS “Catalog” not “Catalogue”
Correction done: “Catalog”

9) Line 187: please state the p-value threshold for the “top signal SNPs”

Correction done. **“Regional plots for the top signal (lowest p-value) that reached suggestive significance of association for BP traits in the Stage 1 and 2 GWASs ($p < 5E-06$) are shown in Figure S 7.”**

10) Lines 218-219, please also provide the REFs here for the datasets you used for PRS

Correction done. References added:

“Polygenic risk scores (PRS), developed from three ancestry (African ¹⁷, European ⁹ and multi-ancestry ¹⁶) GWASs (discovery) were applied to the individuals in AWI-Gen cohort (target, N=10,676) for SBP and DBP (shown in Figure 4).”

11) Please make it clear in your REFs list, if any REFs are pre-prints, e.g. REF 38, perhaps others...

Correction done: “Warren, H. *et al.* Genome-wide analysis in over 1 million individuals reveals over 2,000 independent genetic signals for blood pressure. **10 March 2022, PREPRINT (Version 1) available at Research Square [<https://doi.org/10.21203/rs.3.rs-1409164/v1>]** (2022).”

Reviewer #3

Major

The authors perform a meta-analysis using the Han and Eskin's random effects model. The specific model supposed to increase the power of a study under heterogeneity. However, it has been empirically observed that the specific model, many times, provides over-estimated p-values. I would strongly suggest to run the meta-analysis using conventional fixed effects models, as in the vast majority of performed GWASs. Also, the wording at the last sentence of this paragraph seems kind of confusing, not sure which method the authors compare against. Please, also make sure about the jargon used here. RE2 is not a method, it's just how Metasoft describes the Han and Eskin random-effects for simplicity.

RE2 was described as "Han-Eskin random effects (RE2) model" (Results - Genetic associations with BP traits (and Methods) and not as a method in text.

Justification of the use of RE2 was given in methods with additional information provided:

- Supplementary Note 2.1
- **"There are major differences in the incidence of HTN within the continent. Also, environmental and lifestyle factors, which are key modifiers of the condition have been reported to differ substantially between the study sites. Therefore, we considered it appropriate to allow for some level of heterogeneity in the meta-analysis and focused on RE2 based P-values. This approach assumes no heterogeneity of effect sizes if the null hypothesis is true (i.e. all beta values are zero), thus correcting for the overly conservative standard RE (random-effects) meta-analysis approach. The RE2 model has shown to achieve higher statistical power than FE (fixed-effects), when there is heterogeneity (unlike the traditional RE), indicating that using RE2 improves detection for discovering associations in GWAS meta-analysis¹⁸. For comparison, FE models, using the same parameters as the RE2 model, were conducted."**
- Fixed effects values (beta, SE, P) and additional RE2's values (Mean Effect, Heterogeneity values, I²) were reported in supplementary data for Tables S 5 (Stage 1) and S 6 (Stage 2)
- Included as a main Figure (for better visualization of GW effect sizes, confidence intervals, and summary statistics for each genetic variant analyzed, and heterogeneity):
- **"Figure S 6: Forest plot showing the effect sizes of the novel GW significant associations (p<5E-08) for (a) SBP and (b) PP in the different geographic regions of the AWI-Gen study."**
- Discussion: Compared to GW SNPs
- For SBP: **"This SNP was not GW significant in the fixed effects (FE) model (p=2.58E-05) (Table S 4, Table S 6, see Supplementary Note 2.1) due to variability of effect between region (Figure S 6). Though, indirectly this supports a possible functional connection between the P2RY1 gene and the trait."**
- For PP: **"The rs115808349 SNP also reached GW significance for the FE model (p=1.25E-08) (Table S 6, see Supplementary Note 2.1)."**

Overall the wording in the methods section for stage 1 and 2 is kind of repetitive, try to make this more clear by removing stuff to the supplementary material.

The Authors feel it necessary for stage 1 and 2 to remain in the Methods section, as it is important for the reader to understand how the different Stages were conducted.

The terminology of Stage 1 and 2 might be a bit confusing for the reader. Therefore, the Authors feel it necessary for the stages to remain in the Methods section, as it is important for the reader to understand how the different Stages were conducted.

The stages are repetitive. Repeated wording, such as pipelines used, has been removed from Stage 2 (where possible) to avoid repetition without losing meaning.

I find the design of this study a little bit confusing. So, the researchers perform GWAS and meta-analysis in stage 1 which is fine. Then in the so called stage 2, additional cohorts are added for SBP and DBP only, correct? Therefore, why you choose to present summary results from each stage separately given that for SBP and DBP the sample has been doubled?

The Authors acknowledge that the terminology of Stage 1 and 2 might be a bit confusing for the reader. Stage 2 here means a meta-analysis of our data with external data.

The stage 1 discovery GWAS includes five BP-related traits (SBP, DBP, HTN, PP and MAP). Stage 2 could not include HTN, PP and MAP due to the lack of summary statistics data available for these BP-related traits across the additional cohorts (UKBBa and UGR). Due to regional diversity, Stage 1 and stage 2 did not give the same (or double) GW or suggestive associations as stage 1 for SBP and DBP. Therefore different tables were generated with results compared (Results – Stage 2):

- “Most of these signals were driven by the Uganda Genome Resource (UGR) dataset ($p < 5E-04$, Table S 7). Only one SBP (rs17428471) and five DBP (rs114007149, rs141245590, rs474277, rs617549, rs556594) independent signals were also identified in the Stage 1 GWAS, reaching suggestive significance. The signals with the lowest p-values that reached suggestive significance were: SBP, rs115702999 (ncRNA_exonic *HECW2:AC020571.3*, $p = 2.77E-07$); DBP, rs6009081 (intronic *PPARA*, $p = 5.75E-07$).”

I’m confused regarding the results of fine-mapping provided in the manuscript. Based on methods the authors have used a Bayesian approach but in the reporting of the results it seems that they focus mostly of on the regional locus zoom plots which actually is not fine-mapping unless I’m missing something here.

Figure 3 has been renamed from “Fine-Mapping” to “Regional plots”. Additional analyses were included for Finemapping:

- **Methods - Fine-mapping (addition in bold):**
 - “The H3ABioNet/H3Agwas Finemapping pipeline workflow was used for fine-mapping (<https://github.com/h3abionet/h3agwas/tree/master/finemapping>), **to identify potential causal variants and credible sets (region set at 300kb, using p-value z-scores to re-estimate beta and se). Shogun stochastic search was performed to identify credible sets of potential causal variants, at 95% confidence level, using FINEMAP⁴ which employs Bayesian calculation of posterior probability.**”
- **Included supplementary data:**
 - “**Table S 10: Fine-mapping the region around independent significant SNPs for BP-related traits with GW significance ($p < 5E-08$).**”
- **Results - Fine-mapping and functional analysis:**
 - For SBP: “**Five SNPs from this region were included in the 95% credible set and the lead SNP (rs77846204) also showed highest probability of being the causal SNP ($\log_{bf} > 2$) (Table S 10).**”

- For PP: **“Five SNPs from this region were included in the 95% credible set and the lead SNP (rs115808349) also showed highest probability of being the causal SNP (logbf>2) (Table S 10).”**

Please report the rules for QC of BP traits to assess outliers. Also, the binary outcomes should be removed from the supplementary figure cited.

Winsorise very extreme values were used to check for outliers. The rules for QC of BP traits to assess outliers were reported. The winsorised values were included in Figure S 1. The binary outcomes (HTN status and Sex) were removed from Figure S 1.

For clarification, the following modification was made:

- **Methods - BP measurements:**
- **“QC was performed on the phenotype data using Stata V15 (StataCorp, College Station, Texas, 77845, US) ¹⁹ to assess outliers and distribution (Figure S 1). The Winsorise very extreme value approach was used to assess outliers i.e. the values should be <6 standard deviations (SD) above or below the mean, but no such values were observed (Figure S 1).**
- **Figure S 1 renamed as: “Distribution for phenotypes with continuous data.”**

How were the close relatives of the participants have been identified and excluded?

Relatedness was identified and excluded by using an exclusion criteria, and accounting for relatedness in analyses conducted.

As mentioned under Methods (Study participants):

- **“Exclusion criteria for the study were: pregnant women, **close relatives** of existing participants (first and second-degree relatives), recent immigrants (who migrated <10 years ago into the region) and individuals with physical impairments preventing measurement of BP.”**

As mentioned in Methods Genetic association analysis - Discovery GWAS (Stage 1 AWI-Gen GWAS):

- **“As the participants originate from East, West and Southern Africa, there was significant population structure across regions (Figure S 4); moreover, preliminary analysis indicated relatedness among individuals from some of the AWI-Gen cohorts. Therefore, adjustments based on PCs (addressing genetic population structure) and kinship-matrix (**addressing relatedness**) was used as covariates.”**
- **“Linear mixed models (LMMs) were used to account for fixed and random effects for **relatedness**.”**

Table 2 is huge and not informative as is. Keep only the necessary information if needed and move the rest to the supplementary material. Same for tables 3 and 4, no need to be at the main manuscript.

The reviewer raises a good point, as this table was too long to remain as main table. The following Table modifications were made:

- **Table 2 has been modified to only show the two genome-wide signals and the original Table 2 moved to the supplementary table (now referenced as Table S 4). Unnecessary columns removed in both Table 2 and Table S 4.**
- **Table 3 had unnecessary columns removed (same columns as Table 2). Table 3 moved to the supplementary table (now referenced as Table S 5).**
- **Table 4 was too large for main tables and move to supplementary material (now referenced as Table S 8).**

- There are now two main tables (Table 1-2).

The PRS analysis is not clear. Which pvalue threshold did the authors choose in order to construct their PRSs? They mention deciles in their analyses but they present quartiles in the figures.

For clarity of the P-value threshold (PT), the following was added to Methods:

- **The P-value threshold (PT) was determined in PRSice-2 V2.3.5 7, by calculating the empirical P-value for each PRS (algorithms described in Supplementary Note 3.3). Different PT were identified for each trait and compared using Rsq, where the best PT was defined by the highest Rsq.”**

The correct term should be “quintile” (When a set of data is divided into five equal parts, each of them is called a quintile, which refers to both the cut-off points as well as the group of values contained. When a set of data is divided into ten equal parts, each of them is called a decile). Incorrect use of the terms deciles or quantile score for PRS has been changed to quintile throughout paper were incorrectly mentioned (including Figure 4).

It is highly suggested that the methods should be polished and provide exact information that the reader can go through the manuscript, tables and figures. As it stands right now it seems that in a lot of places, the authors just choose the prespecified thresholds provided from the tools they use without providing this kind of information to the reader.

The Authors agree with the reviewer. Modifications have been made throughout these sections.

- **Methods – Modifications made for clarity of methods used. Removed repeated information and included explanation of methods used. Extra details have been moved to the Supplementary Information document (Supplementary Note 1-3).**
- **Figures and tables – Modifications for clarity of results reported. Main figures and tables were further limited by moving to supplementary information. In addition, other necessary figures and tables were included in the supplementary information.**

Minor

The reported heritability estimates in lines 65-66 are for African populations? Is so, please make it clear to the text. If it is a general statement, the heritability explained for blood pressure traits is now quite larger (line 66). The authors can check a paper that they already cite in their manuscript (Evangelou et al.) that provides larger estimates.

The Authors would like to thank the Reviewer for this suggestion. Since this was a general statement, Evangelou, et al. ⁹ used referenced for larger estimates.

Line 65-66 reworded as: “Genome-wide association studies (GWASs) have explained **27% genetic heritability for BP** ⁹.”

Line 96: I would not call HTN a trait, it’s a medical condition. Please harmonize through out the text.

HTN is defined as a binary trait (corresponding to the presence or absence of HTN) when referring to BP measures. When referring to a person being hypertensive or having HTN the word “binary trait” is not included. For clarification, the definition of HTN was updated to state that: HTN is “**A medical condition commonly known as high BP that occurs when the force of the blood against the artery walls is too high**” (see Table S 1: BP definitions, according to JNC7 guidelines.)

Line 98: Please correct “the” to “that”

Correction made: “that”

Line 523: Please change “ICPBP” to “ICBP”. Same in line 566

Correction made: “ICBP”

Line 227: Please show the pvalues

Correction made: Reference made to **Table S 13**, where $p < P$ -value threshold (PT) values are reported.

References

- 1 Li, Y.-H., Zhang, G.-G. & Wang, N. Systematic characterization and prediction of human hypertension genes. *Hypertension* **69**, 349-355 (2017).
- 2 Liu, C. *et al.* Meta-analysis identifies common and rare variants influencing blood pressure and overlapping with metabolic trait loci. *Nature genetics* **48**, 1162 (2016).
- 3 Sung, Y. J. *et al.* A multi-ancestry genome-wide study incorporating gene–smoking interactions identifies multiple new loci for pulse pressure and mean arterial pressure. *Human molecular genetics* **28**, 2615-2633 (2019).
- 4 Benner, C. *et al.* FINEMAP: efficient variable selection using summary data from genome-wide association studies. *Bioinformatics* **32**, 1493-1501 (2016).
- 5 Robin, X. *et al.* pROC: an open-source package for R and S+ to analyze and compare ROC curves. *BMC bioinformatics* **12**, 1-8 (2011).
- 6 Team, R. C. R: A language and environment for statistical computing. (2013).
- 7 Euesden, J., Lewis, C. M. & O’reilly, P. F. PRSice: polygenic risk score software. *Bioinformatics* **31**, 1466-1468 (2014).
- 8 Warren, H. *et al.* Genome-wide analysis in over 1 million individuals reveals over 2,000 independent genetic signals for blood pressure. **10 March 2022, PREPRINT (Version 1) available at Research Square** [<https://doi.org/10.21203/rs.3.rs-1409164/v1>] (2022).
- 9 Evangelou, E. *et al.* Genetic analysis of over 1 million people identifies 535 new loci associated with blood pressure traits. *Nature genetics* **50**, 1412 (2018).
- 10 Buniello, A. *et al.* The NHGRI-EBI GWAS Catalog of published genome-wide association studies, targeted arrays and summary statistics 2019. *Nucleic acids research* **47**, D1005-D1012 (2019).
- 11 Kamat, M. A. *et al.* PhenoScanner V2: an expanded tool for searching human genotype–phenotype associations. *Bioinformatics* **35**, 4851-4853 (2019).
- 12 Giri, A. *et al.* Trans-ethnic association study of blood pressure determinants in over 750,000 individuals. *Nature genetics* **51**, 51-62 (2019).
- 13 Franceschini, N. *et al.* Genome-wide association analysis of blood-pressure traits in African-ancestry individuals reveals common associated genes in African and non-African populations. *The American Journal of Human Genetics* **93**, 545-554 (2013).
- 14 Hoffmann, T. J. *et al.* Genome-wide association analyses using electronic health records identify new loci influencing blood pressure variation. *Nature genetics* **49**, 54 (2017).
- 15 Feitosa, M. F. *et al.* Novel genetic associations for blood pressure identified via gene-alcohol interaction in up to 570K individuals across multiple ancestries. *PloS one* **13** (2018).
- 16 Wojcik, G. L. *et al.* Genetic analyses of diverse populations improves discovery for complex traits. *Nature* **570**, 514-518 (2019).
- 17 Gurdasani, D. *et al.* Uganda genome resource enables insights into population history and genomic discovery in Africa. *Cell* **179**, 984-1002. e1036 (2019).
- 18 Han, B. & Eskin, E. Random-effects model aimed at discovering associations in meta-analysis of genome-wide association studies. *The American Journal of Human Genetics* **88**, 586-598 (2011).
- 19 StataCorp, L. Stata statistical software: release 15 College Station, TX, 2017. *Erişim Adresi: www.stata.com/features/documentation/(last accessed on 1 March 2018). Erişim Tarihi* **28**, 2022 (2017).

REVIEWERS' COMMENTS

Reviewer #2 (Remarks to the Author):

Well done to the Authors for this thorough Revision. All of my Reviewer comments have been sufficiently addressed, with clear responses. Thank you. I have no further comments. Thank you for your contribution to both the field of BP genetics, and the field of non-EUR genetics.